# HQGS: High-Quality Novel View Synthesis with Gaussian Splatting in Degraded Scenes

**Xin Lin**[1,2,*]  **Shi Luo**[3,*]  **Xiaojun Shan**[1]  **Xiaoyu Zhou**[4]

**Chao Ren**[3]  **Lu Qi**[2]  **Ming-Hsuan Yang**[5]  **Nuno Vasconcelos**[1]

[1]UCSD  [2]Insta360 Research  [3]Sichuan University  [4]Peking University  [5]UC Merced

## Abstract

3D Gaussian Splatting (3DGS) has shown promising results for Novel View Synthesis. However, while it is quite effective when based on high-quality images, its performance declines as image quality degrades, due to lack of resolution, motion blur, noise, compression artifacts, or other factors common in real-world data collection. While some solutions have been proposed for specific types of degradation, general techniques are still missing. To address the problem, we propose a robust HQGS that significantly enhances the 3DGS under various degradation scenarios. We first analyze that 3DGS lacks sufficient attention in some detailed regions in low-quality scenes, leading to the absence of Gaussian primitives in those areas and resulting in loss of detail in the rendered images. To address this issue, we focus on leveraging edge structural information to provide additional guidance for 3DGS, enhancing its robustness. First, we introduce an edge-semantic fusion guidance module that combines rich texture information from high-frequency edge-aware maps with semantic information from images. The fused features serve as prior guidance to capture detailed distribution across different regions, bringing more attention to areas with detailed edge information and allowing for a higher concentration of Gaussian primitives to be assigned to such areas. Additionally, we present a structural cosine similarity loss to complement pixel-level constraints, further improving the quality of the rendered images. Extensive experiments demonstrate that our method offers better robustness and achieves the best results across various degraded scenes. Source code and trained models are publicly available at: https://github.com/linxin0/HQGS.

## 1 Introduction

Novel view synthesis advanced significantly in recent years, with the introduction of Neural Radiance Fields (NeRF) and 3D Gaussian Splatting (3DGS), benefiting applications such as augmented reality (AR) and virtual reality (VR) (Bian et al., 2016; Dawood, 2009; Farshid et al., 2018; Fassi et al., 2016). Nevertheless, existing methods assume high-quality images captured with precise camera parameters. When faced with images of low resolution, with motion blur, compression artifacts, noise, or other degradations common in real-world imaging, they often struggle. Some NeRF (Bahat et al., 2022; Ma et al., 2022; Zhou et al., 2023b; Pearl et al., 2022; Wu et al., 2024) and 3DGS (Feng et al., 2024) variants attempt to address the problem by incorporating various strategies or constraints, such as degradation kernels or super-sampling networks. However, these models are tailored for specific types of degradation and show poor generalization across degradation type and strength, even collapsing for severely degraded imagery. This is illustrated in Figure 1, where NeRFLix and SRGS, two methods designed to address blur and low-resolution scenes, respectively, fail to handle each other's scenarios effectively. When considered over the range of degradation conditions considered in the figure, their performance is not superior to those of the original NeRF and 3DGS models.

Correspondence to Lu Qi: qqlu1992@gmail.com

Xin Lin and Shi Luo share the equal contribution.

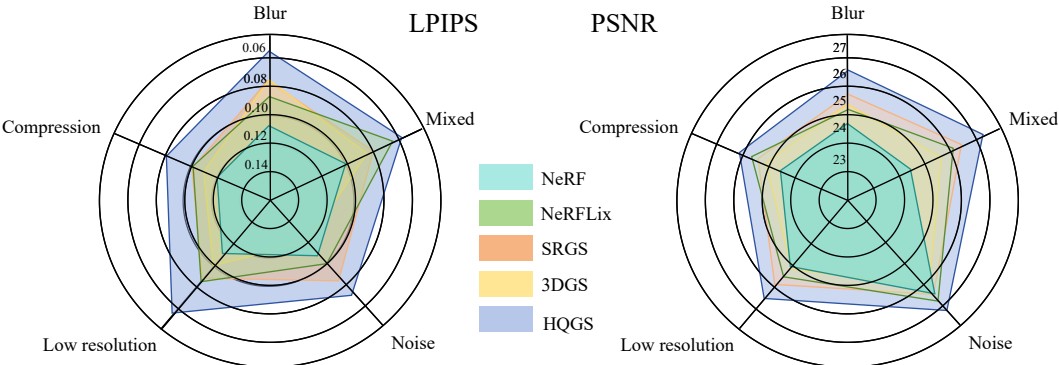

Figure 1: Results of existing methods (Mildenhall et al., 2021; Kerbl et al., 2023; Zhou et al., 2023b; Feng et al., 2024) and our HQGS on five degradation scenes. HQGS performs well against others in these different degradation types.

To improve performance of 3DGS under a range of challenging conditions, we first consider the commonalities among the various types of image degradation of Figure 2 on the stages of 3D reconstruction and rendering. For reconstruction, our preliminary experiments (Figure 2(a)) show that, under degraded conditions, the distribution of recovered Gaussian primitives becomes too sparse to allow the capture of fine scene details, especially for small objects. To address this, we propose an Edge-Semantic Fusion Guidance (ESFG) module that explicitly encourages the reconstruction to direct attention to these details. ESFG is implemented by complementing the stream of scene views with a stream of views that capture the high-frequency image details, computed with an edge detection operator. A pair of semantic-aware and edge-aware feature representations are then extracted from these streams and fused by cross-attention. These features are finally used to refine the parameters of the Gaussian primitives extracted by 3DGS.

For rendering, our preliminary experiments (Figure 2(b)) confirm the findings of (Dong et al., 2023; Lin et al., 2023b), showing that the low-frequency information derived from the global scene structure is essential for high quality rendering. We thus introduce a global Structural Cosine Similarity (SCS) loss to encourage consistency of global (low-frequency) structure between rendered and target images. The ESFG module and SCS loss are finally combined with the 3DGS model to produce a High-Quality Gaussian Splatting (HQGS) rendering model.

Extensive experimental results, under various types of degradation (low resolution, JPEG compression, noise, blur, and mixed degradation), demonstrate the benefits of both the ESFG module and the SCS loss, as well the superiority of HQGS over previous NeRF and 3DGS approaches, for novel view synthesis from degraded imagery. Notably, HQGS performs robustly on more highly degraded images. Overall, this work makes the following contributions:

- We formulate the problem of improving the robustness of novel view image rendering under various types of degradation, including low resolution, noise, blur, JPEG compression, and their combinations.
- We hypothesize that solutions to this problem can benefit from explicitly reasoning in terms of the frequency content of input views, namely by encouraging the rendering model to pay attention to high frequency details during reconstruction and consistency of global (low-frequency) information during rendering.
- We introduce a new framework, HQGS, for the solution of this problem that maps this hypothesis into a pair of mechanisms: the ESFG module, which combines and edge contour prior and semantic information to modulate the sensitivity of the model to finegrained scene details, and the SCS loss, which encourages consistency of global structures between rendered and target images.
- We present evidence from comprehensive experiments confirming the validity of the hypothesis and showing that HQGS has significantly better robustness to image degradation than existing approaches.

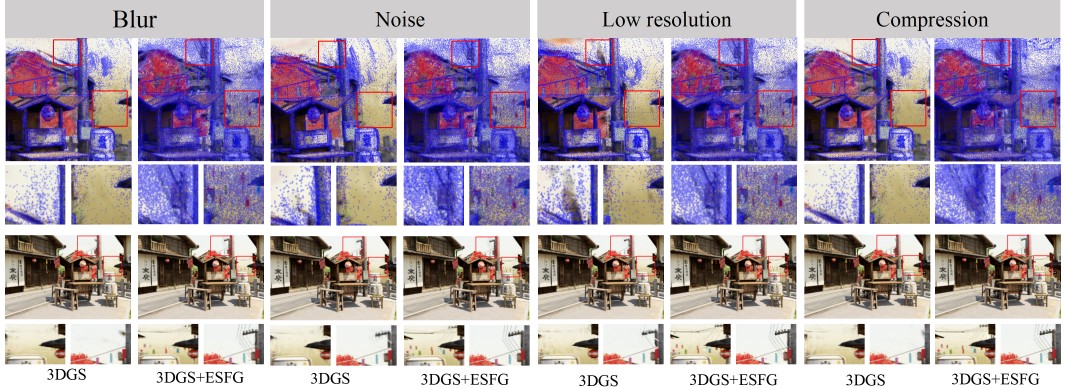

(a) Comparison of Gaussian primitives and rendered images of 3DGS with proposed edge-semantic fusion guidance (ESFG) module under various degraded scenes.



(b) Comparison of the difference map between the rendered image and the clean version, which contains high- and low-frequency information.

Figure 2: (a) The visualization of the distribution of Gaussian primitives in a trained 3DGS under multiple degradation scenes, along with the rendered 2D images. Due to our edge-semantic fusion guidance module providing priors for detailed regions, it can distribute more Gaussian primitives in some detailed areas. As a result, the rendered novel view images contain richer details, particularly in elements like power lines and colorful flags. (b) We compare the difference map between the rendered image from existing methods and its corresponding clean one. A lot of high-frequency details and low-frequency areas involve some structures that are bright, which means these areas are not learned well. In contrast, HQGS achieves better performance in both low-frequency and high-frequency regions.

## 2 RELATED WORKS

**Novel View Synthesis.** Mildenhall et al. (Mildenhall et al., 2021) introduce the neural radiance field (NeRF) to implicitly represent static 3D scenes and synthesize novel views from multiple images with known poses. Building on this foundation, numerous NeRF-based models (Xu et al., 2022; Deng et al., 2022; Chen et al., 2022; Fridovich-Keil et al., 2022; Garbin et al., 2021; Reiser et al., 2021; Chen et al., 2023) have been developed. Point-NeRF (Xu et al., 2022) and DS-NeRF (Deng et al., 2022) integrate sparse 3D point cloud and depth information to resolve the geometric ambiguities inherent in NeRFs, leading to more accurate 3D point sampling and improved rendering quality. On the other hand, Plenoxels (Fridovich-Keil et al., 2022), TensoRF (Chen et al., 2022), FastNeRF (Garbin et al., 2021), KiloNeRF (Reiser et al., 2021), and MobileNeRF (Chen et al., 2023) focus on employing various advanced techniques to accelerate the training and inference processes.

Most recently, 3D-GS (Kerbl et al., 2023), based on point cloud rendering, facilitates real-time novel view synthesis. While these methods have made notable progress in rendering high-quality scenes, they can still produce artifacts with low-quality images and imprecise camera poses.

**Novel View Synthesis in Degraded Scenes.** Several NeRF- and 3DGS-based approaches (Zhou et al., 2025; Huang et al., 2022; Bahat et al., 2022; Feng et al., 2024; Pearl et al., 2022; Zhou et al., 2023b) synthesizing high-quality novel views from degraded scenes using paired clean and degraded images. NeRF-SR (Wang et al., 2022) enhances output resolution through sub-pixel sampling, though this approach demands more computational resources and extends training times. On the other hand, NVSR (Bahat et al., 2022) trains a NeRF super-resolution (SR) network with multi-view data, leveraging the triplane structure to perform SR on low-resolution planes, thereby improv-

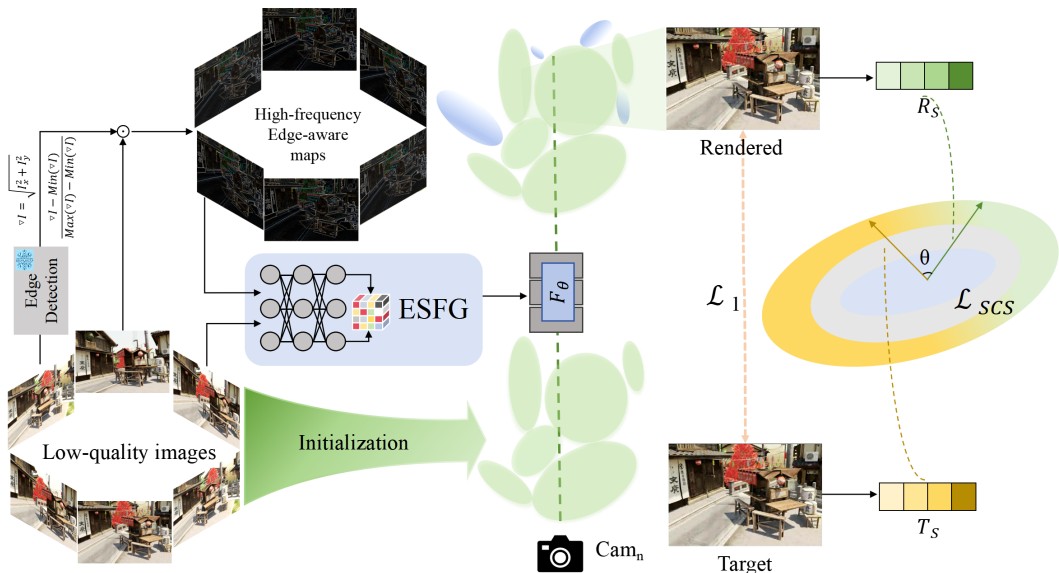

Figure 3: HQGS framewok. The high-frequency edge-aware maps and low-quality images are jointly fed into the learnable ESFG module for feature modulation and fusion. The fused features guide the training process of HQGS, providing reliable detail priors. During training, we employ pixel-level $\mathcal{L}_1$ loss and a global structural cosine similarity loss $\mathcal{L}_{SCS}$ to optimize the model.

ing the overall NeRF resolution. In (Huang et al., 2022), RefSR-NeRF introduces a specialized SR module that incorporates high-resolution reference images, which can lead to longer training and inference times. To achieve high-quality rendering results in the noisy scene, NaN (Pearl et al., 2022) adapts the feed-forward IBRNet view synthesis method to achieve competitive burst denoising results. Instead of focusing on 3D learning, NeRFLiX (Zhou et al., 2023b) and NeRFLiX++ (Zhou et al., 2023a) learn a general 2D viewpoint mixer via simulated image degradation. However, if the distribution of the rendering artifacts shifts from the simulated data, the performance degrades. Drantal-NeRF (Yang et al., 2024) employs a diffusion-based image quality enhancement model to create higher-quality image pairs for training a NeRF model. Recently, SRGS (Feng et al., 2024) has emerged as a 3DGS-based model capable of achieving super-resolution rendering using paired low- and high-quality images.

**Image Quality Enhancement and Restoration.** Image restoration aims to enhance the quality of degraded images affected by various types and levels of degradation. This challenging task encompasses denoising (Lin et al., 2023b; Ren et al., 2021; 2022), deblurring (Fang et al., 2023; Sun et al., 2023; Pan et al., 2023), and general restoration (Zamir et al., 2021; 2022a; Lin et al., 2023a; 2024). Restormer (Zamir et al., 2022a) incorporates transformers for low-level restoration to balance the performance and computational costs, and MIRNetv2 (Zamir et al., 2022b) restores images through a novel feature extraction network. On the other hand, Kong et al. (Kong et al., 2023) present a Transformer-based method for high-quality image deblurring, by utilizing frequency-domain properties to simplify scaled dot product attention and alleviate the need for complex matrix multiplication. Recently, MRLPFNet (Dong et al., 2023) proposes a simple yet efficient low-pass filtering network, and demonstrate the importance of repairing low-frequency regions to improve image quality. SCPGabNet (Lin et al., 2023b) and SCP$^2$GAN (Lin et al., 2024) are non-paired unsupervised restoration methods that exploit the importance of low-frequency constraints and introduce background consistency loss (BGM) specifically for low-frequency areas.

## 3    PROPOSED METHOD

Figure 3 shows the overall pipeline of the proposed HQGS that receives multi-view low-quality images $I \in \mathcal{R}^{N \times H \times W \times 3}$ as the input. The $N$, $H$ and $W$ are the number, height and width of images. Using these images, we first generate the initial point cloud and corresponding camera

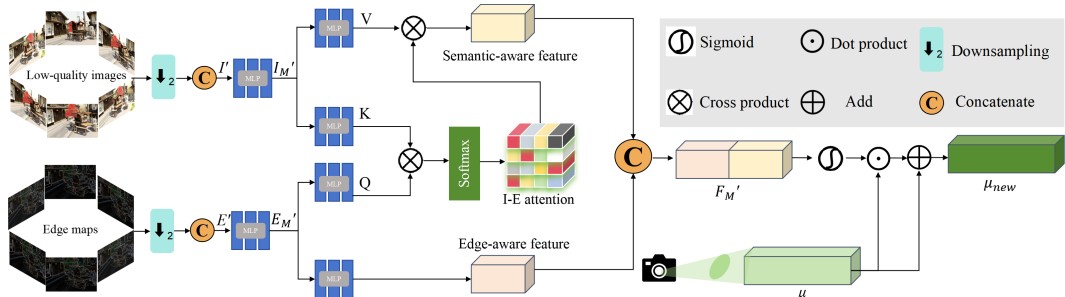

Figure 4: The ESFG module separately learns semantic-aware feature and edge-aware feature, and then jointly guides the training of HQGS.

views through the classical COLMAP method. Meanwhile, we detect the boundaries of each image to construct the high-frequency edge-aware maps. Next, we train a structure-assisted 3DGS model to leverage structural information in both the reconstruction and rendering stages. On the one hand, we propose an edge-semantic fusion guidance (ESFG) module that utilizes high-frequency edge-aware maps as a condition signal for reconstructing 3DGS. On the other hand, we utilize structural cosine similarity loss ($\mathcal{L}_{SCS}$) to constrain the global structure when rendering Gaussian primitives into an image within a view.

In the following subsections, we first introduce the various degradation types designed in our method. Then, we will elaborate on the ESFG module and $\mathcal{L}_{SCS}$, for robust HQGS.

## 3.1 DEGRADATION TYPES

To account for the range of adverse conditions found in real-world conditions, we consider five types of image degradation. These are simulated by applying the following operations to the sets of high-quality images commonly used in the literature:

- *Low Resolution:* $4\times$ downsampling is applied to create low-resolution images.
- *JPEG Compression:* Images are compressed using JPEG with setting the quality level 10.
- *Blur:* Blurred images are generated using a Gaussian blur kernel with a radius ranging from 10 to 20, and the blur angle is randomly selected within the range $[0, 2\pi]$.
- *Noise:* Gaussian noise with a standard deviation 10 is added to the images.
- *Mixed Degradation:* A combination of the degradations above, applied in the following order: low resolution, JPEG compression, blurring, and noise.

Following 3DGS (Kerbl et al., 2023), we input a set of low-quality scene images with the corresponding cameras calibrated by COLMAP, producing a sparse point cloud as a side effect. Figure 2(a) compares low-quality and high-quality scenes with their corresponding point clouds. The low-quality images provide less visual information, creating a sparser point cloud than the high-quality scene.

## 3.2 EDGE-SEMANTIC FUSION GUIDANCE MODULE

**Motivation.** Since degradation affects the edge details of some objects in the scene, making them difficult to capture. Figure 2(a) shows Gaussian primitives distributed in these areas, such as power lines and colorful flags, become sparse. This causes these objects to be omitted in the rendered 2D images, reducing the quality of the rendered novel views. To address this problem, we leverage specific edge contour priors as a 'reminder' to inform the model that particular objects in these regions must be generated. This encourages it to cover these areas with more Gaussian primitives during reconstruction.

**Solution.** We propose the Edge-Semantic Fusion Guidance (ESFG) module to enhance semantics from low-quality images by the high-frequency edge-aware maps. To extract global high-frequency edge-aware maps ($E$), we use the Sobel operator to calculate the gradient maps $\nabla I \in \mathcal{R}^{N \times H \times W \times 3}$

of the low-quality images $I$ that contain critical edge information. We then normalize $\nabla I$ to the gradient masks $\nabla I'$ along with height and width dimensions:

$$\nabla I' = \frac{\nabla I - Min(\nabla I)}{Max(\nabla I) - Min(\nabla I)}, \tag{1}$$

where $Max(\cdot)$ and $Min(\cdot)$ is the maximum and minimum value in $\nabla I$. Lastly, we obtain the high-frequency edge-aware map $E$ by

$$E = \nabla I' \odot I. \tag{2}$$

where $\odot$ represents matrix multiplication. The $E \in \mathcal{R}^{N \times H \times W \times 3}$ highlights rich high-frequency image textures and details crucial for image quality enhancement and novel view synthesis. The difference between $\nabla I'$ and $E$ is presented in the appendix.

In the ESFG module, we down-sample them to $I'$ and $E'$ by $2\times$ as the input for computational efficiency where $I'$ and $E' \in \mathcal{R}^{N \times \frac{H}{2} \times \frac{W}{2} \times 3}$. Then, we concatenate the images of $I'$ and $E'$ according to the camera order, respectively. To match the position parameters $\boldsymbol{\mu} \in \mathcal{R}^{M \times 3}$ of the Gaussian primitives, we perform a scale transformation to obtain $I'_M$ and $E'_M \in \mathcal{R}^{\frac{M}{2} \times 3}$ using MLPs, where $M$ represents the number of Gaussian primitives.

The ESFG module contains two branches in Figure 4. In the upper branch, we employ a cross-attention mechanism where $E'_M$ acts as the query, and the $I'_M$ serves as the key and value. This cross-attention between the low-quality images and the high-frequency edge-aware maps connects local high-frequency details with global semantics, producing more comprehensive guided features, i.e., semantic-aware features. In the lower branch, a learnable MLP layer is used to better interpret edge information, resulting in the edge-aware feature. The output features from both branches are then concatenated to form the final fused feature $F'_M$, followed by a sigmoid non-linear operation. Finally, $F'_M$ is used to modulate the original position parameters $\boldsymbol{\mu} \in \mathbb{R}^3$ of the Gaussian primitives to obtain new position parameters $\mu_{new}$, and then HQGS models it as $G(x)$:

$$\mu_{new} = Sigmoid(F'_M) \odot \mu + \mu, \tag{3}$$

$$G(x) = e^{(-\frac{1}{2}(x - \mu_{new})^T \Sigma^{-1}(x - \mu_{new}))}. \tag{4}$$

where $\odot$ represents matrix multiplication, and $\boldsymbol{\Sigma} \in \mathbb{R}^{3 \times 3}$ is an anisotropic covariance matrix, which is factorized into a scaling matrix $\boldsymbol{S}$ and a rotation matrix $\boldsymbol{R}$ as $\boldsymbol{\Sigma} = \boldsymbol{R}\boldsymbol{S}\boldsymbol{S}^\top \boldsymbol{R}^\top$.

## 3.3 STRUCTURAL COSINE SIMILARITY LOSS

While high-frequency information is essential for enhancing image quality, low-frequency information should not be overlooked, as it corresponds to smooth areas and colors closely related to global structural information (Dong et al., 2023; Lin et al., 2023b).

We propose a Structural Cosine Similarity Loss ($\mathcal{L}_{SCS}$) to enhance the global structure of low-frequency regions and improve overall rendering quality. This loss emphasizes directional consistency in the low-frequency feature space, allowing it to better capture the image's global structure and thereby achieve a closer match to its overall characteristics.

In the optimization stage, besides the original $\mathcal{L}_1$ between the rendered image ($R$) and its corresponding target image ($T$). We add our $\mathcal{L}_{SCS}$ as a low-frequency-aware loss. Specifically, we first obtain the low-frequency structure-aware map (S) of the $R$ and $T$:

$$R_S = (1 - \nabla I') \odot R, \quad T_S = (1 - \nabla I') \odot T. \tag{5}$$

We then compute structural cosine similarity loss $\mathcal{L}_{SCS}$:

$$\mathcal{L}_{SCS} = 1 - \frac{\sum_{i=1}^{N} R_S^i \cdot T_S^i}{\|R_S\|_2 \cdot \|T_S\|_2}, \tag{6}$$

where $N$ is the total number of pixels, and $i$ represents the $i$-th pixel. The overall loss function is:

$$\mathcal{L} = \lambda_1 \mathcal{L}_1 + \lambda_2 \mathcal{L}_{SCS}, \tag{7}$$

where $\lambda_1$ and $\lambda_2$ denote weight parameters.

| | Methods | Low resolution | | Compression | | Motion Blurry | | Gaussian Noisy | | Mixed | | Ren time (s) |
|---|---|---|---|---|---|---|---|---|---|---|---|---|---|
| | | PSNR↑ | LPIPS↓ | PSNR↑ | LPIPS↓ | PSNR↑ | LPIPS↓ | PSNR↑ | LPIPS↓ | PSNR↑ | LPIPS↓ | |
| NeRF based | NeRF | 25.02 | 0.106 | 24.72 | 0.122 | 24.78 | 0.109 | 26.33 | 0.115 | 24.18 | 0.102 | ~4.7 |
| | Nan | 23.73 | 0.257 | 20.75 | 0.349 | 21.92 | 0.192 | 21.27 | 0.476 | 20.97 | 0.541 | ~10.7 |
| | NVSR | 25.31 | 0.117 | 25.42 | 0.157 | 25.13 | 0.097 | 26.52 | 0.129 | 25.61 | 0.096 | ~4.3 |
| | NeRFLiX | 25.58 | 0.093 | 25.71 | 0.112 | 25.53 | 0.088 | 26.69 | 0.097 | 26.04 | 0.064 | ~7.5 |
| 3DGS based | 3DGS | 25.19 | 0.096 | 25.31 | 0.113 | 25.68 | 0.075 | 26.20 | 0.119 | 24.88 | 0.085 | ~0.2 |
| | SRGS | 25.96 | 0.087 | 25.54 | 0.091 | 25.92 | 0.076 | 26.47 | 0.086 | 26.30 | 0.074 | ~0.2 |
| | **HQGS** | 26.48 | 0.065 | 26.19 | 0.077 | 26.55 | 0.056 | 27.20 | 0.071 | 27.27 | 0.057 | ~0.2 |

Table 1: Results of both NeRF-based and 3DGS-based methods on LLFF dataset (Ma et al., 2022). Ren time denotes the rendering time for each frame. The red color indicates the best results, and the blue color indicates the second-best results.

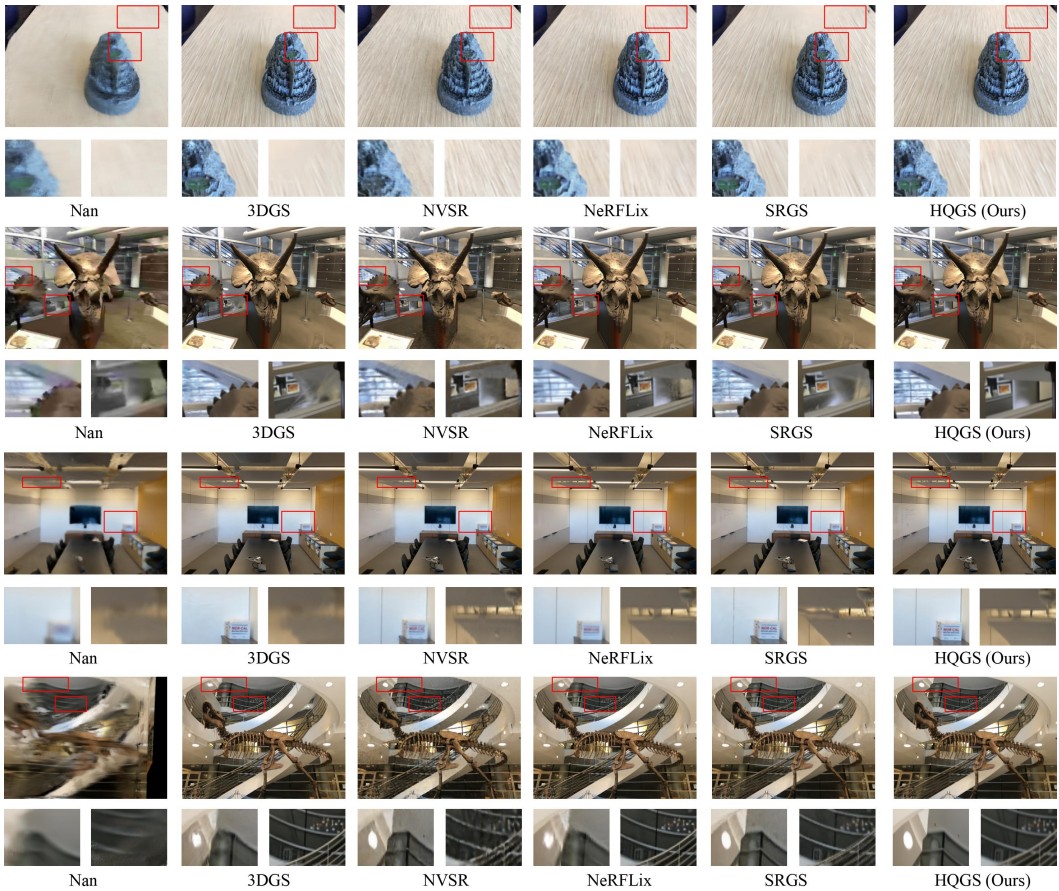

Figure 5: Visualization of the low-resolution, compression, blurry and noisy scenes on LLFF (Mildenhall et al., 2019).

# 4 EXPERIMENTS

We first describe the utilized datasets and present the implementation details. Next, we provide a comprehensive analysis of the experimental results, qualitatively and quantitatively. Finally, we conduct ablation studies to validate the effectiveness of the proposed modules and the robustness of our HQGS compared to other approaches.

## 4.1 EXPERIMENTAL SETUP

**Datasets.** We evaluate the proposed HQGS pipeline on two datasets: (1) The LLFF dataset (Mildenhall et al., 2019), which contains real-world images from eight distinct scenes, with each scene comprising 20 to 62 images. Of these, 1/4 are reserved for testing, while the remaining are used for training. (2) A synthetic dataset derived from the Blender scenes used in DeblurNeRF (Ma et al., 2022), where 1/8 of the data is used for testing and the other 7/8 for training.

**Implementation Details.** Our implementation is based on the 3DGS (Kerbl et al., 2023) framework. The learning rate for the learnable parameters of 3D Gaussians follows the official settings, while the learning rate for the ESFG module is set to 1e-6. We evaluate our method using various metrics, including PSNR, SSIM, and LPIPS, following previous work (Zhou et al., 2023c; Kerbl et al., 2023; Feng et al., 2024). All experiments are conducted on a single Nvidia GeForce RTX 3090 GPU. Following previous methods, we use the trained restoration network to get the target image for the results comparisons. The $\lambda_1$ and $\lambda_2$ in equation 7 are set to 1 and 5, respectively.

**Evaluated Methods.** We evaluate methods that provide code and pre-trained models for fair and comprehensive comparisons. For NeRF-based methods, we choose NeRF (Mildenhall et al., 2021), Nan (Pearl et al., 2022), NVSR (Bahat et al., 2022), and NeRFLiX (Zhou et al., 2023b). In terms of 3DGS-based approaches, 3DGS (Kerbl et al., 2023) and SRGS (Feng et al., 2024) are used for comparison. We retrain all methods using low and restored high-quality pairs. The pre-trained IVM is used in NeRFLiX. To comprehensively compare the efficiency and performance of the networks, all 3DGS-based models are trained for 50,000 iterations. Meanwhile, we also provide a comparison of the training time in the following section.

## 4.2 RESULTS AND COMPARISONS.

**Novel View Reconstruction on LLFF (Mildenhall et al., 2019).** Table 1 compares HQGS to all NeRF (Mildenhall et al., 2021; Pearl et al., 2022; Bahat et al., 2022; Zhou et al., 2023b) and 3DGS (Kerbl et al., 2023; Feng et al., 2024) baselines on the LLFF dataset, showing that HQGS achieves the best results across all degradation conditions. In the low-resolution setting, it achieves gains of 1.46 dB/0.041, 0.90 dB/0.028, 1.29 dB/0.031, and 0.52 dB/0.022 in PSNR/LPIPS over NeRF, NeRFLiX, 3DGS, and SRGS, respectively. For JPEG compression conditions, it has gained 0.48 dB/0.035 and 0.65 dB/0.014 over NeRFLiX and SRGS. It can also be seen that NeRF-based methods, such as NeRFLiX, require 7.5 seconds for rendering vs. only 200 milliseconds for HQGS. Figure 5 compares images synthesized by the different methods under low resolution, compression, blurry, and noisy conditions. The images generated by HQGS contain richer high-frequency details and more precise visual effects.

**Novel View Reconstruction on DeblurNeRF (Ma et al., 2022).** Table 2 presents the average PSNR and LPIPS of different methods under five degradation conditions. HQGS achieves the best performance under all metrics. Compared to the NeRFLix (Zhou et al., 2023b), it has a gain of 0.79 dB/0.008/0.016 on PSNR/SSIM/LPIPS. HQGS also delivers substantial improvements across all metrics over SRGS (Feng et al., 2024), which is designed for resolution enhancement.

| Methods | PSNR↑ | SSIM ↑ | LPIPS↓ |
|---|---|---|---|
| NeRF | 26.16 | 0.894 | 0.067 |
| Nan | 21.87 | 0.738 | 0.105 |
| NVSR | 26.45 | 0.895 | 0.072 |
| NeRFLiX | 26.95 | 0.896 | 0.061 |
| 3DGS | 26.27 | 0.895 | 0.072 |
| SRGS | 26.82 | 0.898 | 0.063 |
| **HQGS (Ours)** | 27.74 | 0.904 | 0.045 |

Table 2: Results on DeblurNeRF dataset (Ma et al., 2022).

## 4.3 ABLATION STUDIES ON THE PROPOSED MODULES.

In this section, we conduct ablation studies on the proposed ESFG module and $\mathcal{L}_{SCS}$. We use the ground truth as the target image. Additional analyses can be found in the appendix.

**Effectiveness of the ESFG Module.** In Table 3, we validate the effectiveness of the proposed ESFG module. V1 represents the original 3DGS. V2 extends V1 by incorporating a semantic-aware feature (SAF) from low-quality images, while V3 uses edge-aware features (EAF) extracted from

| Methods | SAF | EAF | Concat | CA | PSNR (dB)↑ | SSIM↑ | LPIPS↓ |
|---------|-----|-----|--------|-----|-----------|-------|--------|
| V1 | | | | | 27.63 | 0.893 | 0.067 |
| V2 | ✓ | | | | $28.05^{\uparrow 0.42}$ | $0.895^{\uparrow 0.002}$ | $0.066^{\downarrow 0.001}$ |
| V3 | | ✓ | | | $28.61^{\uparrow 0.98}$ | $0.905^{\uparrow 0.012}$ | $0.058^{\downarrow 0.009}$ |
| V4 | ✓ | | | ✓ | $28.42^{\uparrow 0.79}$ | $0.901^{\uparrow 0.008}$ | $0.062^{\downarrow 0.005}$ |
| V5 | ✓ | ✓ | ✓ | | $28.92^{\uparrow 1.29}$ | $0.913^{\uparrow 0.020}$ | $0.052^{\downarrow 0.015}$ |
| V6 | ✓ | ✓ | ✓ | ✓ | $29.26^{\uparrow 1.63}$ | $0.914^{\uparrow 0.021}$ | $0.049^{\downarrow 0.018}$ |

Table 3: Effectiveness of the proposed ESFG module.

| Methods | $\mathcal{L}_1$ | $\mathcal{L}_{BGM}$ | $\mathcal{L}_{SP}$ | $\mathcal{L}_{SCS}$ | PSNR (dB)↑ | SSIM↑ | LPIPS↓ |
|---------|-----|-----|-----|-----|-----------|-------|--------|
| V1 | ✓ | | | | 29.26 | 0.914 | 0.049 |
| V2 | ✓ | ✓ | | | $29.42^{\uparrow 0.16}$ | $0.916^{\uparrow 0.002}$ | $0.045^{\downarrow 0.004}$ |
| V3 | ✓ | | ✓ | | $29.39^{\uparrow 0.13}$ | $0.917^{\uparrow 0.003}$ | $0.043^{\downarrow 0.006}$ |
| V4 | ✓ | | | ✓ | $29.91^{\uparrow 0.65}$ | $0.919^{\uparrow 0.005}$ | $0.031^{\downarrow 0.018}$ |

Table 4: Ablation studies on the proposed $\mathcal{L}_{SCS}$. The experiments are conducted on the 'Wine' scene with blurry degradation from the DeblurNeRF dataset.

| Method | 3DGS | 3DGS | 3DGS | 3DGS+ESFG | 3DGS+ESFG | 3DGS+ESFG |
|--------|------|------|------|-----------|-----------|-----------|
| Sobel | ✓ | | | ✓ | | |
| Gaussian | | ✓ | | | ✓ | |
| Laplace | | | ✓ | | | ✓ |
| PSNR(dB)↑ | 28.61 | 28.27 | 28.41 | 29.26 | 28.86 | 28.95 |
| SSIM↑ | 0.905 | 0.902 | 0.904 | 0.914 | 0.908 | 0.908 |
| LPIPS↓ | 0.058 | 0.066 | 0.063 | 0.049 | 0.056 | 0.055 |

Table 5: Ablation studies on different guidance. The experiments are conducted on 'Wine' scenes with blurry degradation from the DeblurNeRF dataset.

high-frequency edge-aware maps to guide 3DGS. V4 enhances the framework by applying cross-attention (CA) between SAF and EAF. V5 fuses SAF and EAF by concatenating, and V6 represents our complete framework with all four components. The specific structures of these variants are shown in detail in the appendix.

Using SAF slightly improves PSNR from 27.63 dB to 28.05 dB. Replacing SAF with EAF (V3) improves PSNR by 0.56 dB compared to V2. However, V4 with the cross-attention mechanism (CA) further leads to a 0.37 dB improvement over V2. Combining both SAF and EAF results in a 0.31 dB/0.008/0.006 and 0.87 dB/0.018/0.014 gain in PSNR/SSIM/LPIPS over using SAF or EAF individually. Finally, incorporating all components (V6) achieves the best results, with a 1.63 dB/0.021/0.018 improvement in PSNR/SSIM/LPIPS over V1.

**Effectiveness of $\mathcal{L}_{\mathbf{SCS}}$.** We assess the effectiveness of the $\mathcal{L}_{SCS}$ in Table 4. V1 denotes the 3DGS with the ESFG module trained using only $\mathcal{L}_1$. V2 builds upon V1 by incorporating a structure-aware pixel-level loss ($\mathcal{L}_{SP}$), which calculates the $\mathcal{L}_1$ between structural elements in the rendered and target images. V3 introduces the $\mathcal{L}_{BGM}$ (Lin et al., 2023b), and V4 replaces the $\mathcal{L}_{SP}$ with our structural cosine similarity loss ($\mathcal{L}_{SCS}$). We observe that adding the $\mathcal{L}_{BGM}$ to the $\mathcal{L}_1$ provides a slight improvement (approximately 0.1 dB in PSNR). Similarly, $\mathcal{L}_{SP}$ results in a marginal improvement. However, substituting local structural constraints with the global $\mathcal{L}_{SCS}$ yields a significant 0.65 dB improvement over the original L1 loss, indicating that global low-frequency structural information is more effective at capturing overall image consistency than pixel-level constraints.

**Comparison of Other Methods for Obtaining High-Frequency Maps.** We compare the results of different operators in both the 3DGS framework (V3 in Table 3) and the 3DGS with our ESFG module (V6 in Table 3). The visualization of these maps is shown in the appendix. Table 5 shows that high-frequency information from the Gaussian filter and Laplace operator can improve performance. However, it is still lower than our high-frequency edge-aware feature in both V3 and V6 frameworks.

Table 6: Robustness validation on two progressive degradation test sets on (Ma et al., 2022). The red color indicates the best results, and the blue color indicates the second-best results.

| | Gaussian noise | | | | Low resolution | | | |
| --- | --- | --- | --- | --- | --- | --- | --- | --- |
| Methods | 0 | 10 | 25 | 50 | 1 × | 2 × | 4 × | 8 × |
| | PSNR LPIPS | PSNR LPIPS | PSNR LPIPS | PSNR LPIPS | PSNR LPIPS | PSNR LPIPS | PSNR LPIPS | PSNR LPIPS |
| NeRF | 30.42 0.072 | 29.11 0.079 | 26.78 0.091 | 23.04 0.143 | 29.83 0.113 | 29.71 0.128 | 28.83 0.138 | 28.16 0.148 |
| 3DGS | 30.21 0.043 | 29.19 0.052 | 27.46 0.077 | 22.78 0.109 | 30.25 0.091 | 30.18 0.094 | 29.25 0.092 | 28.63 0.111 |
| NeRFLix | 30.86 0.054 | 29.47 0.063 | 27.04 0.071 | 23.32 0.112 | 31.42 0.069 | 30.87 0.069 | 30.12 0.076 | 29.66 0.096 |
| SRGS | 30.71 0.036 | 29.17 0.045 | 27.63 0.061 | 23.21 0.126 | 31.36 0.078 | 30.93 0.065 | 30.54 0.061 | 30.11 0.087 |
| **HQGS** | 31.32 0.018 | 30.41 0.029 | 28.63 0.043 | 26.31 0.067 | 32.08 0.031 | 31.85 0.033 | 31.61 0.038 | 31.37 0.051 |

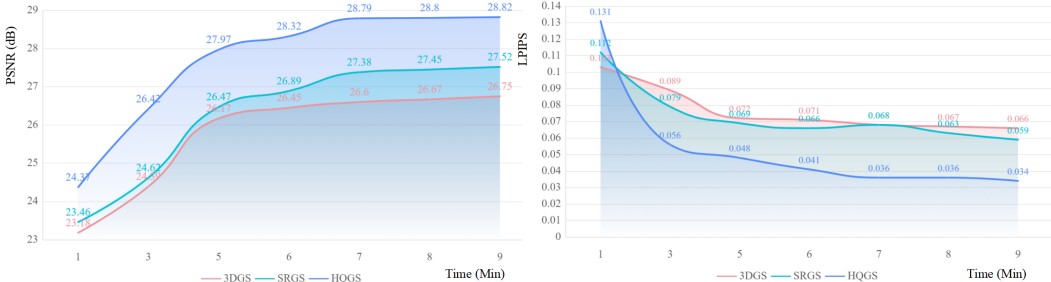

Figure 6: Training time vs quality for blurry scenes. HQGS performs well against 3GDS and SRGS under both the PSNR and LPIPS metrics.

## 4.4 ANALYSIS ON ROBUSTNESS OF OUR HQGS.

So far, we have considered mild image degradations. In this section, we investigate the robustness of HQGS across degradation levels. We construct two types of progressive degradation test sets: (1) Noise: Gaussian noise with variances of 0 (clean), 10, 25, and 50, and (2) Low resolution: images downsampled to 1× (clean), 2×, 4×, and 8× scales. Table 6 shows that the gains of HQGS over existing methods increases with the degradation strength and are quite large for the strongest degradations for both types of degradation. Especially as noise becomes severe (variance increasing from 25 to 50), all methods exhibit a sharp drop, for example, SRGS from 27.63 dB to 23.21 dB and NeRFLix from 27.04 dB to 23.32 dB, making high-quality rendering challenging. In contrast, HQGS has a much smaller decline and maintains a robust performance of 26.31 dB, which achieves a 3.10 dB and 2.99 dB improvement compared to SRGS and NeRFLix. A similar trend is observed in low-resolution scenarios, where our method consistently demonstrates more robust performance than others. For instance, under the 8 × setting, our method achieves a 1.26 dB/0.036 and 1.71 dB/0.045 increase over SRGS and NeRFLix on PSNR and LPIPS.

## 4.5 TRAINING TIME VS QUALITY.

In general, there is a trade-off between rendering quality and time, as higher quality rendering can be achieved by considering more views of the scene. Figure 6 compares the performance of different 3DGS-based approaches as a function of training time. It is clear that HQGS achieves a better trade-off than 3DGS and SRGS. For instance, a 5-minute reconstruction by HQGS has a 1.22 dB (0.018) gain in PSNR (LPIPS) over the 3DGS results for 9 minutes.

## 5 CONCLUSION

In this paper, we propose a robust Gaussian variant HQGS, which performs favorably under various degradation conditions and exhibits strong robustness as the degradation level increases. Building upon the limitations of the existing 3DGS in rendering scene details due to degradation, we propose an edge-semantic fusion guidance module to guide the positional parameters of 3DGS. Furthermore, during optimization, we introduce a global structural cosine similarity loss ($\mathcal{L}_{SCS}$) to complement the pixel-level $\mathcal{L}_1$ loss. Extensive experimental results demonstrate the effectiveness and robustness of the proposed HQGS.

**Acknowledgements.** This work was partially funded by NSF award IIS-2303153.

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

## A    SPECIFIC STRUCTURE OF VARIANTS IN SECTION 4.3 OF THE MAIN PAPER

Five comparative variants are tested to verify the influence of the proposed Edge Fusion Guidance (ESFG) module. Since the experimental results have been reported in sub-section 4.3, we mainly describe their specific structures in this section. As the V1 in Table 3 of the main paper is the 3DGS without learnable modules, we only provide details for the remaining variants. As shown in Figures. 7, 8, 9 and 10.

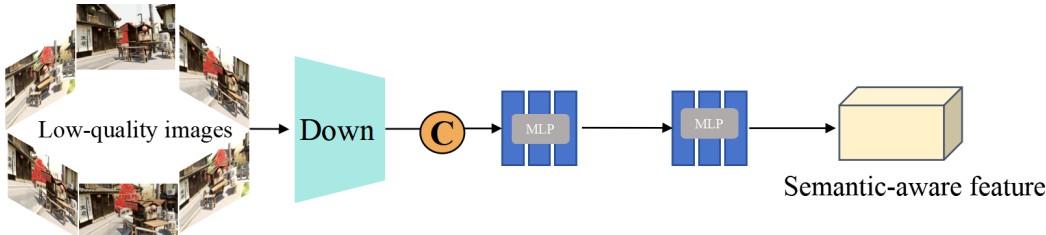

Figure 7: The structure of V2 in Table 3 of the main paper.

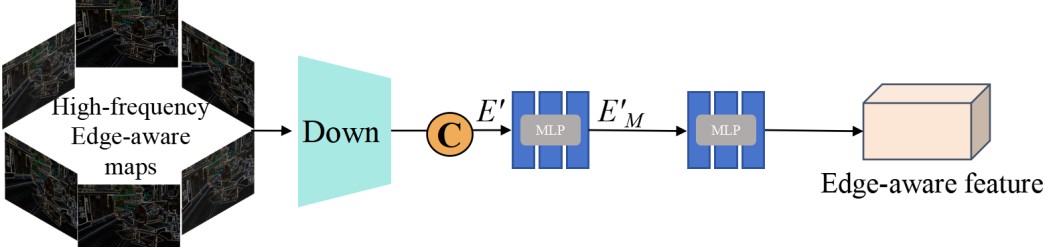

Figure 8: The structure of V3 in Table 3 of the main paper.

## B    VISUALIZATION RESULTS OF OTHER METHODS FOR OBTAINING HIGH-FREQUENCY MAPS.

Our goal is to capture contour regions representing the image's high-frequency content. We first show the $\nabla I'$ and $E$ in Section 3.2 in the main paper in Figure 11. After that, as shown in Figure 12, we explore other methods for extracting high-frequency content, such as using a Gaussian high-pass filter and Laplace operator to obtain high-frequency responses.

## C    ROBUSTNESS VALIDATION ON EDGE MAPS.

We leverage the well-known fact (T., 2009) that edges have sufficient frequency information and can be obtained by an edge detection operator even from degraded images. We provide visualizations of edge detection results obtained using the Sobel operator under progressively challenging conditions, as shown in Figure 13 in the submitted Supplementary Material. Additionally, we quantified the results by calculating the PSNR metrics of the images and edge maps under various conditions compared to the clean ones. Then, we presented the PSNR variance of the images and edge maps: 10.18/0.285. This further demonstrates that the edge maps are more robust under progressively degraded conditions. Thus, edge detection performs well in extracting edge information to a considerable extent, even under relatively challenging conditions.

## D    ABLATION STUDIES ON THE DOWNSAMPLING PARAMETERS IN THE ESFG MODULE.

We introduce downsampling in the ESFG module to reduce dimensionality and save computational resources. To further investigate its impact, we supplement additional experiments under two con-

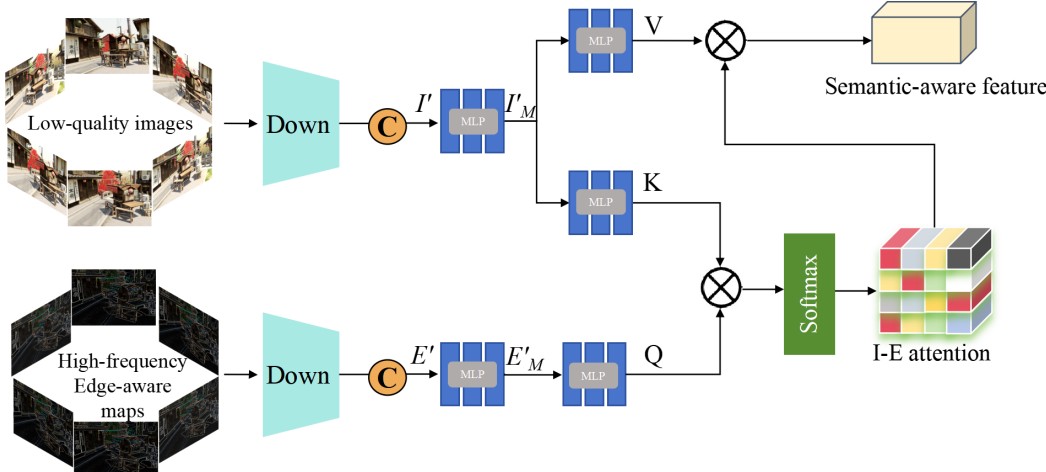

Figure 9: The structure of V4 in Table 3 of the main paper.

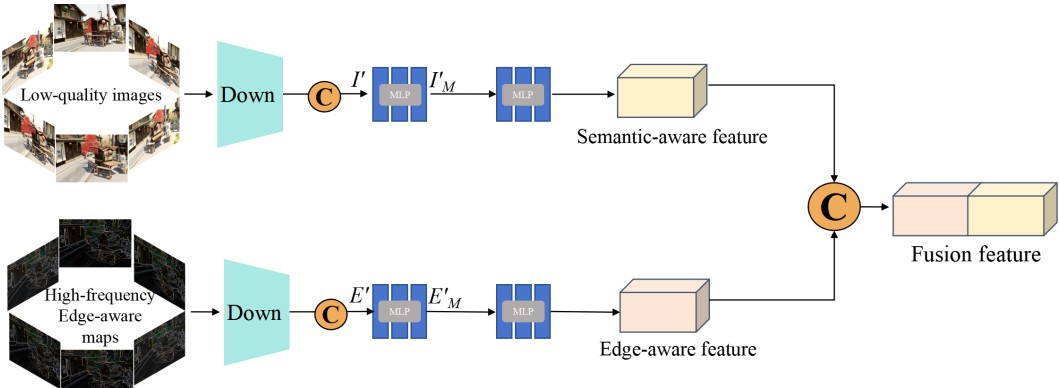

Figure 10: The structure of V5 in Table 3 of the main paper.

figurations: Without Downsampling and $4 \times$ Downsampling in Table 7. Therefore, we choose the $2 \times$ to balance the training time and the performance.

# E    ADDITIONAL QUALITATIVE RESULTS.

This section presents additional results to underscore the advanced capabilities of our HQGS over other leading NeRF-based and 3DGS-based techniques. Illustrated in Figures. 14, our methods consistently deliver superior quality of rendered image.

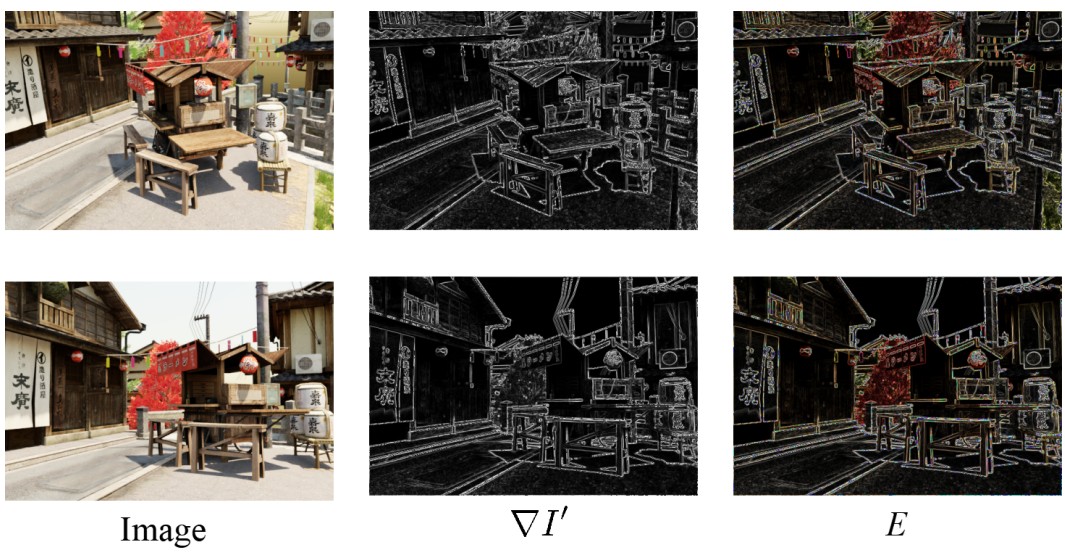

Figure 11: Visualization the $\nabla I'$ and $E$.

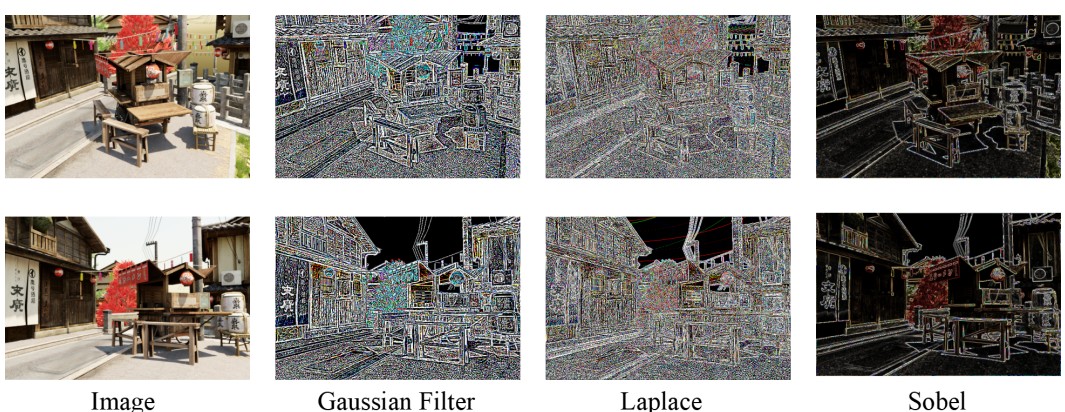

Figure 12: Visualization the map from different operators.

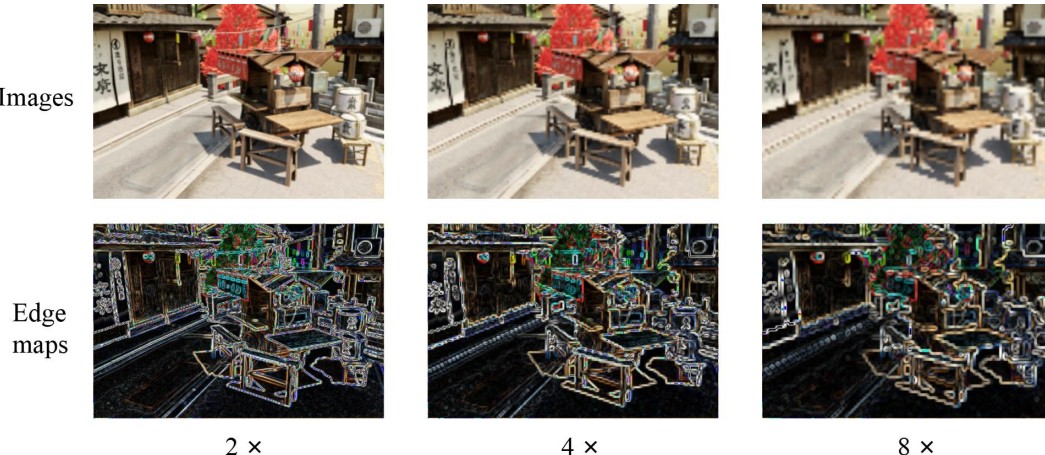

Figure 13: Visualization of edge maps on progressive degradation scenes.

| Methods | $1 \times$ | $2 \times$ | $4 \times$ |
|---------|------------|------------|------------|
| PSNR(dB)↑ | 31.79 | 31.70 | 31.42 |
| Time(Min)↓ | 16.4 | 13.3 | 11.2 |

Table 7: Ablation Studies on the Downsampling Parameters in the ESFG Module.

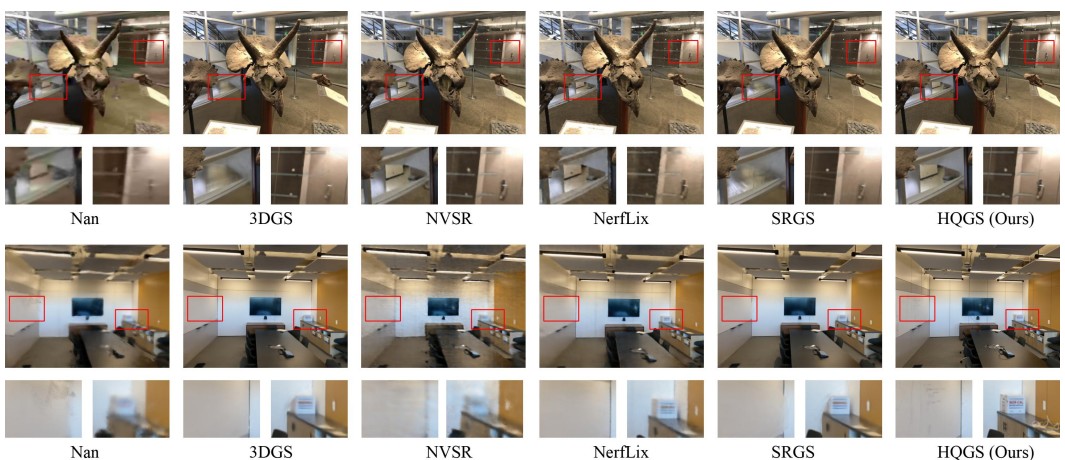

Figure 14: Visualization results on the LLFF dataset.

