# OpenReview forum: "HQGS: High-Quality Novel View Synthesis with Gaussian Splatting in Degraded Scenes"
_ICLR.cc/2025/Conference — ICLR 2025 Poster_

### Official Review · Reviewer_enTq · 2024-10-22

**Soundness:** 3
**Presentation:** 3
**Contribution:** 3
**Rating:** 6
**Confidence:** 4

**Summary:**

The authors proposed a novel training strategy that can help to reconstruct 3D scenes from low-quality images.

**Strengths:**

The proposed method performs better than other compared SOTAs on 3D reconstruction form low-quality images. And the idea that learning a modulation for the position is good.

**Weaknesses:**

I think the description of the paper is not clear. Please see my following questions.

**Questions:**

1. How to apply the colmap on low quality images. I think it's not accurate.
2. What is M in line 265.
3. Why do you downsample the I and E by 2?
4. In Eqn. 3, authors used F'M, while in the above contents, authors used F'. What's the difference of them?
5. No other layers after the fusion features but before the sigmoid?
6. I guess the M represents the number of points, then how to get fusion features in dimension M?
7. In the original 3DGS, there is a loss called D-SSIM loss. Does it help to emphasizes directional consistency in the low-frequency feature space? Why do you change it with SCS loss?
8. The number of points is not fixed. 3DGS will split and clone points. How to you know how many M do you need?

---

> ### Author Response · Authors · 2024-11-22
> **Response to Reviewer enTq**
>
> We thank reviewer enTq for acknowledging the contribution of our paper and providing thoughtful comments.
>
> **Q1. Explanation of the COLMAP Section.**
>
> We follow the setup in 3DGS [1], which states, "the input to our method is a set of images of a static scene, together with the corresponding cameras calibrated by SfM （COLMAP）, which produces a sparse point cloud as a side effect. From these points, we create a set of 3D Gaussians." In our case, we replace the clean images with low-quality images to generate the corresponding point cloud by COLMAP. We make it clear in Sec. 3.1 of the revised manuscript.
>
> [1] Bernhard Kerbl, Georgios Kopanas, Thomas Leimkuhler and George Drettakis. 3d gaussian splatting for real-time radiance field rendering. ACM Trans. Graph. 2023: 1–14.
>
> ---
>
> **Q2, 6, 8. Explanation of M and ESFG.**
>
>
> We clarify in the revised manuscript that M represents the number of Gaussian primitives. As you mentioned, with the split-and-clone process, the value of M dynamically changes.
>
>
> The ESFG output is a fused feature that contains global information of the scene. To handle the changes in M, we fix the scale of the fused feature and replicate it according to the varying number of Gaussian primitives (M). This approach ensures that the fused feature consistently retains global information while avoiding the time costs of adjusting dimensions through learnable parameters.
>
> ---
>
> **Q3. Explanation of the Downsampling Parameters in the ESFG Module.**
>
> As mentioned in line 263 in the original manuscript, we introduce downsampling in the ESFG module to reduce dimensionality and save computational resources. To further investigate its impact, we supplement additional experiments under two configurations: Without Downsampling and 4 $\times$ Downsampling. Last, we choose the 2 $\times$ to balance the training time and the performance.
>
> | Method   | 1 $\times$ | 2 $\times$ | 4 $\times$|
> |----------|------|------|------|
> | **PSNR(dB)↑** | 31.79 | 31.70 | 31.42 |
> | **Time(Min)↓**     | 16.4 | 13.3 | 11.2 |
>
> ---
>
> **Q4. Some Typos.**
>
> Thanks for your advice! The paper has been revised to enhance the writing quality.
>
> ---
>
> **Q5. Are there layers after the fusion features but before the sigmoid?**
>
> There are no additional learnable parameters between the fusion features and the sigmoid activation. All learnable parameters are placed prior to the fusion feature generation process.
>
>
> ---
>
> **Q7. Ablation Studies on the Proposed Loss.**
>
> As you suggested, we add the ablation studies on D-SSIM and our 𝓛₍SCS₎. In the following Table, our 𝓛₍SCS₎ have better results. For 3D reconstruction in degraded scenes, our framework first leverages edge maps to enhance the perception of high-frequency object edges. Simultaneously, the proposed 𝓛₍SCS₎ focuses more on the low-frequency global structure, forming a complementary relationship guided by edges for improved results. In contrast, D-SSIM is not specifically designed as a loss function for low-frequency components. And it does not effectively complement our high-frequency edge perception strategy, resulting in relatively limited performance.
>
>
>
> | Methods | 𝓛₁ | 𝓛₍D-SSIM₎ | 𝓛₍SCS₎ | PSNR (dB)↑    | LPIPS↓     |
> |---------|-----|--------|--------|---------------|------------|
> | V1      | ✓   |       |        | 28.22         | 0.045      |
> | V2      | ✓   | ✓      |      | 28.57 **↑0.35** | 0.038 **↓0.007** |
> | V3      | ✓   |        | ✓      |29.09 **↑0.78** | 0.031 **↓0.014** |

---

> ### Author Response · Authors · 2024-11-24
> **Follow-Up on Rebuttal Discussion**
>
> Dear Reviewer enTq,
>
> We deeply appreciate your valuable feedback during the first round of review and the thoughtful discussion that has significantly helped us refine our work. Since the discussion phase ends on Nov 26, we would like to know whether we have addressed all the issues, and we would greatly welcome any additional feedback or suggestions you may have.
>
> Thank you again for your devotion to the review. If all the concerns have been successfully addressed, please consider raising the scores after this discussion phase.
>
> Best regards,
>
> Paper4233 Authors

---

> > ### Comment · Reviewer_enTq · 2024-11-25
> >
> > In question 1, I am not sure if the images are in low quality, you can use colmap to get the correct pose estimation.

---

> ### Author Response · Authors · 2024-11-25
> **Response to Reviewer enTq**
>
> Thank you for your thoughtful feedback and for taking the time to review our rebuttal. Both high-quality and low-quality images need to be fed into COLMAP to obtain estimated poses and initialized point clouds for 3DGS-based methods. As you suggested, we add a new visualisation of the poses for both high- and low-quality scenes in Figure 4 of the submitted Supplementary Material. These are two scenes from DeblurNeRF and LLFF, respectively, where the red cones represent the cameras. Since the scenes in DeblurNeRF are smaller, the cameras are also smaller, while those in LLFF are larger. We can observe that in low-quality scenarios, the camera's perspective shifts, and the decline in image quality affects the accuracy of pose estimation in COLMAP, resulting in poses that differ from those derived from high-quality images. This has been analyzed in other NeRF-based papers [1], which focus on posing issues to address blurry scenes.
>
> However, our paper, along with some contemporary 3DGS-based studies [2,3,4], emphasises the sparsity of the point cloud, which is one of the key motivations behind our proposed method. Low-quality scenes significantly contribute to the sparsity of the initialized point cloud. Due to the limited information in low-quality images, the initialized point cloud contains significantly fewer points and is much sparser, as shown in Figure 2(a) of our previous and revised manuscript. Additionally, we have provided the number of points in the initialized point clouds in Figure 4 of the submitted Supplementary Material. The points in the point cloud decrease from 32,338 to 11,021 and from 6,174 to 3,348 in high-quality to low-quality scenes across the two datasets.
>
> Nonetheless, all comparison models are retrained in the same setting, ensuring the fairness of the comparisons.
>
> [1] Li Ma, Xiaoyu Li, Jing Liao, Jue Wang, Qi Zhang, Xuan Wang, Pedro V. Sander. Deblur-nerf: Neural radiance fields from blurry images. CVPR. 2022: 12861-12870.
>
> [2] Xiang Feng, Yongbo He, Yubo Wang, Yan Yang, Wen Li, Yifei Chen, Zhenzhong Kuang, Jiajun Ding, Jianping Fan, and Jun Yu. Srgs: Super-resolution 3D Gaussian Splatting. arXiv:2404.10318, 2024
>
> [3] Byeonghyeon Lee, Howoong Lee, Xiangyu Sun, and Eunbyung Park. Deblurring 3d gaussian splatting. European Conference on Computer Vision. Springer, Cham, 2025: 127-143.
>
> [4] Shiyun Xie, Zhiru Wang, Yinghao Zhu, and Chengwei Pan. SuperGS: Super-Resolution 3D Gaussian Splatting via Latent Feature Field and Gradient-guided Splitting. arXiv preprint arXiv:2410.02571, 2024.

---

> > ### Comment · Reviewer_enTq · 2024-11-25
> >
> > I think the reviewer solved all my concerns. I agree to accept this paper.

---

> > > ### Author Response · Authors · 2024-11-26
> > > **Official Comment by Authors**
> > >
> > > We sincerely thank the reviewer for their valuable additional comments and feedback. It is good to see that our comments could address your concerns. Since the concerns have been addressed and you agree to accept this paper, we kindly request if it would be possible to check and increase the score accordingly (as the score remained the same after your last comments). We greatly appreciate your time and consideration.

---

### Official Review · Reviewer_s4XL · 2024-10-27

**Soundness:** 4
**Presentation:** 4
**Contribution:** 4
**Rating:** 8
**Confidence:** 4

**Summary:**

This work is concerned with the improvement of 3D Gaussian Splatting-based radiance fields computed for images that have quality issues. In particular, blur, reduced resolution, compression artifacts, and noise. The authors present a proposed method with two key modifications over the prior art. The first modification is an edge fusion guidance module that merges semantic information with edge information to favor the representation of fine details in the final radiance field overcoming issues with the above distortions. The second key modification is the introduction of a structural cosine similarity loss that acts on the low frequency areas of the rendered images to ensure better representation of low texture areas of the radiance field.

**Strengths:**

This paper is very well written. The authors supply sufficient detail on the proposed method to allow it to be correctly understood both on it's own and in the context of the prior art. The contribution is quite novel and although the edge fusion guidance module is motivated by the prior art, it is certainly not a trivial increment on the prior art and represents a new way of looking at the problem of low quality input images to radiance field training. The experimental section is quite strong, with comprehensive comparisons to the prior art and convincing improvements. The ablation studies are quite thorough, showing that the authors have put a lot of thought into the study and gone to considerable efforts to explore the work. The results of the ablation studies support the inclusion of each aspect of the two proposed modifications clearly. Conclusions are well-founded and justified by the experimental results.

**Weaknesses:**

Overall the work is strong, but there are a couple of areas of improvement. I found that certain figures contained unnecessary details or were difficult to read, while certain aspects of the explanations are unclear or seem contradictory. I also found that the analysis of compression artifacts was somewhat limited. In the "Questions" section of this review, I list these areas specifically and make suggestions for improvements.

**Questions:**

Q1: In the abstract of the work, the author's state: "The fused features serve as prior guidance to capture detailed distribution across different regions, bringing more attention to areas with a higher concentration of Gaussian primitives.". I find this sentence to be confusing. Later in the work it becomes clear that the ESFG module emphasizes edge related information in the input images in order to adapt the layout and properties of Gaussian's to better capture key information during training. In other words, the ESFG module guides the training of the Gaussian primitives by bringing more attention to key areas of the input images. It does not bring "more attention to areas with a higher concentration of Gaussian primitives." as this implies that the ESFG module is concerned with drawing attention to the density of Gaussians in the radiance field, which is not the case. It draws attention to key features of the input images and this in turn effects the density of the Gaussian primitives. I suspect this is what the authors meant, but the language is vague and admits the other interpretation. I suggest the following rewording of this sentence: "The fused features serve as prior guidance to capture detailed distribution across different regions, bringing more attention to areas with higher semantic meaning, such as edges, in turn allowing for higher concentration of Gaussian primitives to be assigned to such areas.".
Q2: On line 48, the authors state "Our preliminary experiments (Figure 2(b)) show that, for reconstruction, the distribution of reconstructed Gaussian primitives becomes too sparse to allow the capture of fine scene details. " Which distortions are the authors referring to here? Noise? Low resolution? Blur? Compression artifacts? All distortions? Please clarify what is being referred in this text? Please state clearly whether this observation refers to specific types of distortions (for example if it refers solely to blur) or whether this statement refers to all types of distortions.
Q3: Figure 4 has a spelling error, "Position paprameter in Gaussians" should be  "Position parameter in Gaussians". In addition, in the caption, the authors state "It separately learns semantic-aware feature and edge-aware feature, and
then jointly guides the training of HQGS." Please avoid the usage of vague terms like "It". What is "It" precisely? For example a potential better sentence is: "The ESFG module learns semantic-aware features and edge-aware features, and...".
Q4: Equation (2) introduces a notation for matrix multiplication that is not explained until after equation 4. Please explain notation at the point at which it is introduced.
Q5: Line 275, "then HQGS model it as G(x)" should be "then HQGS models it as G(x)".
Q6: Line 352, "methods that provide codes and" should be "methods that provide code and".
Q7: Figure 7 is a pastel, set of 3D overlapping bars with partial transparency that make the plot overly artistic and hard to read. A simple set of non-overlapping groups bars would have provided the same information and been clearer.
Q8: Figure 8 contains pastel colored, semi-transparent overlaid plots with some form of fill gradient transitions. The pastel colors are very similar and hard to differentiate in the plot. Please simplify and remove the unnecessary additional graphics. Key information like the numbers on the graphs are overlapping making them difficult to read.
Q9: In section 3.1, the authors state that the JPEG Compression will only be studied at a quality level of 10. Please explain why this particular value was chosen and why only a singular value was chosen for this parameter. In addition, only one value of Low Resolution was selected (4x downsampling). Why was this number chosen? Please provide additional text to describe the justification of the choice of JPEG quality level and downsampling factor. In addition please consider the testing of a wider range of these parameters (for example JPEG quality settings higher and lower than 10 as well as downsampling factors of 2x and 8x). If it is not appropriate to test a wider variety of values for JPEG Compression and downsampling, please state the rationale clearly.

---

> ### Author Response · Authors · 2024-11-22
> **Response to Reviewer s4XL**
>
> We thank reviewer s4XL for acknowledging the contribution of our paper and providing thoughtful comments.
>
> >**Q1, 3, 4, 5, 6, 7, 8. Improving and Rewriting Some Sentences.**
>
> Thanks for your valuable suggestions! We have made several improvements to the writing quality in the revised manuscript. The key changes include:
>
>
> 1. The introduction of ESFG in the abstract has been revised: "The fused features serve as prior guidance to capture detailed distribution across different regions, bringing more attention to areas with detailed edge information and allowing for a higher concentration of Gaussian primitives to be assigned to such areas."
> 2. Figure 4 has been updated, and its caption revised to: "The ESFG module separately learns semantic-aware feature and edge-aware feature, and then jointly guides the training of HQGS."
> 3. The explanation of the notation for matrix multiplication in Equation 2 has been improved for clarity.
> 4. Several sentences have been enhanced for better clarity and readability.
> 5. Figures 7 and 8 from the original manuscript have been modified: Figure 7 has been expanded with additional comparison algorithms, and it is now presented in table format as Table 6 in the revised manuscript. In Figure 8, we have redrawn the figure with more distinguishable colors and adjusted the fonts to eliminate overlapping issues.
>
> ---
>
> >**Q2. Further Explanation on the Distortion in Line 48.**
>
>
> As we initially explore the common effects of various degradation scenarios on the 3DGS model, the experiments involve multiple scenarios. Thus, the term "distortion" here refers to the impact across all these blur, noise, low resolution, and compression scenarios. In the revised manuscript, we add further clarification on this point.
>
> ---
>
> >**Q9. Regarding selecting JPEG Compression Quality Parameters and Additional Experiments under Different Settings.**
>
> We follow several image restoration methods [1, 2, 3], where the JPEG compression quality factors are typically chosen within the range of 10–40. For our experiments, we select a factor of 10, representing the most severe scenario. Additionally, as you suggested, we add experiments under other quality factors, as well as tests for 2 $\times$ and 8 $\times$ downsampling scenarios.
>
>
> |  | **JPEG Compression** 5 | **JPEG Compression** 5 | **JPEG Compression** 10  |**JPEG Compression** 10  | **JPEG Compression** 20  | **JPEG Compression** 20 | **Low resolution** 2×| **Low resolution** 2×| **Low resolution** 4×  |**Low resolution** 4×  | **Low resolution** 8× |     **Low resolution** 8×   |
> |-|-|-|-|-|-|-|-|-|-|-|-|-|
> |  **Methods** | **PSNR↑**| **LPIPS↓** | **PSNR↑** | **LPIPS↓** | **PSNR↑** | **LPIPS↓** | **PSNR↑** | **LPIPS↓** | **PSNR↑** | **LPIPS↓**  | **PSNR↑** | **LPIPS↓** |
> | **NeRF** | 25.32 | 0.146 | 26.83 | 0.129 | 27.67 | 0.113 | 29.71 | 0.128  | 28.83 | 0.138 | 28.16 | 0.148 |
> | **3DGS** | 26.27 | 0.084 | 27.95 | 0.076 | 28.34 | 0.068 | 30.18 | 0.094   | 29.25 | 0.092 | 28.63 | 0.111 |
> | **NeRFLix** | 26.98 | 0.076 | 28.40 | 0.069 | 29.01 | 0.056 | 30.87 | 0.069 | 30.12 | 0.076 | 29.66 | 0.096 |
> | **SRGS** | 27.18 | 0.065 | 28.22 | 0.087 | 28.97 | 0.064 | 30.93 | 0.065 | 30.54 | 0.061 | 30.11 | 0.087 |
> | **HQGS (Ours)** | **27.83** | **0.058** | **28.92** | **0.044** | **29.67** | **0.035** | **31.85** | **0.033** | **31.61** | **0.038** | **31.37**  | **0.051** |
>
>
>
> [1] Li Y, Fan Y, Xiang X, et al. Efficient and explicit modelling of image hierarchies for image restoration. CVPR. 2023: 18278-18289.
>
> [2] Simon Welker, Henry N. Chapman, Timo Gerkmann. DriftRec: Adapting diffusion models to blind JPEG restoration. IEEE Transactions on Image Processing, 2024: 2795-2807.
>
>
> [3] Li B, Li X, Lu Y, et al. PromptCIR: Blind Compressed Image Restoration with Prompt Learning[J]. arXiv:2404.17433, 2024.

---

> ### Author Response · Authors · 2024-11-24
> **Follow-Up on Rebuttal Discussion**
>
> Dear Reviewer s4XL,
>
> We deeply appreciate your valuable feedback during the first round of review and the thoughtful discussion that has significantly helped us refine our work. Since the discussion phase ends on Nov 26, we would like to know whether we have addressed all the issues, and we would greatly welcome any additional feedback or suggestions you may have.
>
> Thank you again for your devotion to the review. If all the concerns have been successfully addressed, please consider raising the scores after this discussion phase.
>
> Best regards,
>
> Paper4233 Authors

---

> > ### Comment · Reviewer_s4XL · 2024-11-24
> > **Reply to Rebuttal**
> >
> > Dear Authors,
> >
> > Thank you to the authors for carefully considering and addressing my comments. I believe that my comments have been fully addressed and support the inclusion of this work in the program.
> >
> > Reviewer s4XL

---

> ### Author Response · Authors · 2024-11-25
> **Official Comment by Authors**
>
> Thank you for your insightful comments and appreciation of our work and rebuttal. It is good to see that our comments could address your concerns. We will do our best to improve the final version of our paper based on your valuable suggestions.

---

### Official Review · Reviewer_cE6C · 2024-10-31

**Soundness:** 2
**Presentation:** 2
**Contribution:** 3
**Rating:** 6
**Confidence:** 3

**Summary:**

The authors identify that 3DGS performs poorly with low-quality images due to insufficient attention to detailed regions, leading to a lack of Gaussian primitives and loss of detail.

To improve this, this paper presents an approach named HQGS, including Edge-Semantic Fusion Guidance Module and Structural Cosine Similarity Loss.

Edge-Semantic Fusion Guidance Module: Combines high-frequency edge-aware maps with semantic information to guide the distribution of Gaussian primitives, enhancing detail in rendered images.

Structural Cosine Similarity Loss: Complements pixel-level constraints by focusing on structural similarities, further improving image quality.

Experimental results demonstrate that HQGS enhances robustness and performance in various degraded scenes.

**Strengths:**

The proposed HQGS framework effectively addresses the challenges of degraded images in novel view synthesis by introducing an Edge-Semantic Fusion Guidance (ESFG) module and a Structural Cosine Similarity Loss (LSCS).

The ESFG module enhances the distribution of Gaussian primitives and improves detail generation, while LSCS ensures global low-frequency structure consistency, leading to higher quality rendered images.

Extensive experiments demonstrate superior robustness and performance in various degradation scenarios, outperforming state-of-the-art methods.

**Weaknesses:**

The method relies heavily on high-quality edge and semantic information, which may be challenging to obtain in extremely degraded or noisy images.

The computational complexity introduced by the ESFG module and LSCS could increase training and inference times, potentially limiting real-time applications.

The presentation of the paper is not optimal in several aspects:
Figure 1 suffers from color blending issues, making it difficult to distinguish between different color regions corresponding to various methods.
Figure 2 is mentioned before Figure 1 in the text, which can be confusing for readers.
Tables 1 and 2 present similar results but use different formatting (one with colored text and one without), leading to inconsistency and potential confusion.
For Figure 5, the effectiveness of the method cannot be understood due to the lack of visualized input views.

**Questions:**

How were the experiments for Table 3 and Table 4 conducted? In which scenes were they performed? What types of degradation were used?

From the input in Figure 2a, it is impossible to see the presence of the "power lines". I am curious whether it is really possible to reconstruct the clear "power lines" in Figure 2b from such low-quality input views. How can this phenomenon be explained? Shouldn't 3D Gaussians be unable to imagine and reconstruct features that are not present (or almost completely blurred) in the input views?

I noticed that the model was trained for 50,000 iterations, which is more than the number used for vanilla 3D-GS. Would this have an impact? If the model is trained for 50,000 iterations, would all other parameters remain unchanged, including those for densification? If so, do the additional 30,000+ iterations seem redundant, or are they used to mainly for the optimization of the MLPs?

Are the weights of the MLP optimized individually for each scene, or are they generalized after pre-training?

Regarding lines 531-532, since you have added an MLP and trained for 50,000 iterations, the training time for HQGS would at least be longer, right?

---

> ### Author Response · Authors · 2024-11-22
> **Response to Reviewer cE6C (Part 1/2)**
>
> We thank reviewer cE6C for acknowledging the contribution of our paper and providing thoughtful comments.
>
> >**Q1. Lack of Robustness Experiments on Edge Maps**
>
>
> We leverage the well-known fact that edge maps have sufficient frequency information [1] and can be obtained by an edge detection operator even from degraded images. As suggested, we visualize edge detection results obtained from the Sobel operator under progressively challenging conditions in Figure 1 of the submitted Supplementary Material. Additionally, we compute the PSNR values based on images and edge maps under various conditions with respect to clean ones, then present the PSNR variance of the images and edge maps: 10.18/0.285. These results further demonstrate that the edge maps are more robust under progressively degraded conditions. Thus, edge detection performs well in extracting edge information to a considerable extent, even under relatively challenging conditions.
>
> In our experimental setups, we follow the degradation settings commonly used in image restoration tasks [2, 3, 4, 5], ensuring that the chosen degradation ranges align with existing works (4$\times$ downsampling, JPEG with quality level 10, and so on). Under these settings, our method consistently performs favorably against existing algorithms. Furthermore, we discuss the progressively challenging scenarios in Figure 7 of the original manuscript, showing that our method performs better than other approaches under severe conditions (e.g., noise level of 50 or 8 $\times$ downsampling). We include comparisons with more methods in Table 6 of the revised manuscript.
>
> [1] Lindeberg T. Scale space. Encyclopedia of Computer Science and Engineering. 2009: 2495–2504.
>
> [2] Li Y, Fan Y, Xiang X, et al. Efficient and explicit modelling of image hierarchies for image restoration. CVPR. 2023: 18278-18289.
>
> [3] Ren B, Li Y, Mehta N, et al. The ninth NTIRE 2024 efficient super-resolution challenge report. CVPR. 2024: 6595-6631.
>
> [4] Lu Z, Li J, Liu H, et al. Transformer for single image super-resolution. CVPR. 2022: 457-466.
>
> [5] El Helou M, Süsstrunk S. Blind universal Bayesian image denoising with Gaussian noise level learning. TIP. 2020, 29: 4885-4897.
>
> ---
>
> >**Q2. Training and inference time.**
>
> In 3D reconstruction, rendering time corresponds to inference time. As shown in Table 2 of the original manuscript (Table 1 of the revised manuscript), we report the rendering times of various methods. Our HQGS achieves comparable rendering times to other 3DGS-based methods and is significantly faster than NeRF-based methods.
>
> Regarding training time, Figure 8 in the original manuscript compares existing methods under the same training time for fairness. HQGS demonstrates a faster convergence rate and consistently performs well against other methods, highlighting its training efficiency. Whether compared under the same training time or at convergence with the same number of iterations, our method shows better performance. To enhance clarity, we have updated the subsection title from "ANALYSIS ON RECONSTRUCTION TIME OF SOME 3DGS-BASED METHODS." to "TRAINING TIME VS QUALITY." and improved the Figure's readability in the revised manuscript to eliminate overlapping issues.
>
>
>
> ---
>
> >**Q3. Some Figures and Tables Need to be Improved.**
>
> Thank you for your valuable feedback! We have revised the manuscript to improve the quality of figures and tables, as detailed below:
>
>
> 1. The colors in Figure 1 have been adjusted to enhance distinguishability and readability.
>
> 2. For the order of the Figures, Figure 1 is referenced on line 42, and Figure 2 on line 48 in the original manuscript.
>
>
> 3. The formatting of Tables 1 and 2 in the original manuscript has been updated for consistency and improved readability.
>
> 4. Regarding the visualized input views, since rendering unseen views in 3D reconstruction only requires the given camera viewpoint information and not the input images. As suggested, the clean ground truth used for PSNR calculation is provided in Figure 5 of the revised manuscript.
>
> ---
>
> >**Q4. The Settings of Experiments in Table 3 and Table 4.**
>
> Thank you for the reminder. Both Table 3 and Table 4 analyze the 'Wine' scene with blurry degradation from the DeblurNeRF dataset. We have updated their titles in the revised manuscript to reflect this context.
>
> ---

---

> > ### Comment · Reviewer_cE6C · 2024-11-30
> >
> > Dear Authors,
> >
> > I would like to appreciate for the comprehensive feedback you have provided. Thank you for the additional clarifications that have undoubtedly enriched the discussion and strengthened the overall contribution of your work.
> > The feedback has addressed most of my confusion. However, I have one more question.  Previous methods for 3DGS often perform ablation studies across multiple scenarios in an entire dataset to mitigate the interference caused by random errors. Is the ablation study conducted in only one scenario(the ‘Wine’ scene with blurry degradation from the DeblurNeRF dataset) sufficient to demonstrate the effectiveness of each module?

---

> > > ### Author Response · Authors · 2024-11-30
> > > **Response to Reviewer cE6C**
> > >
> > > Thank you for your thoughtful feedback and for taking the time to review our rebuttal. We will address your points regarding the ablation studies below and include these in the revised version to enrich the experiment part.
> > >
> > > >**Previous methods for 3DGS often perform ablation studies across multiple scenarios in an entire dataset to mitigate the interference caused by random errors. Is the ablation study conducted in only one scenario(the ‘Wine’ scene with blurry degradation from the DeblurNeRF dataset) sufficient to demonstrate the effectiveness of each module?**
> > >
> > > Our novel view reconstruction performance in Table 1 and Table 2 of the revised version is conducted across multiple scenes following the settings of other 3DGS-based methods. Based on your suggestion, we perform ablation studies on all five scenes—Factory, Cozyroom, Pool, Tanabata, and Trolley (Wine)—from the DeblurNeRF dataset and average the results to validate the effectiveness of the proposed modules. We show the results in the following table, demonstrating the effectiveness of the proposed ESFG and L(SCS) across multiple scenes. We will replace the original ablation study on the Wine scene with the following new table in the revised version.
> > >
> > >
> > >
> > > | Method   | SAF | EAF |Concat | CA |  L(SCS) | **PSNR(dB)↑** | **SSIM↑** | **LPIPS↓**|
> > > |----------|------|------|------|------|------|------|------|------|
> > > | V1 | |  |  |  |  | 27.63 | 0.893 | 0.067 |
> > > | V2 |✓|  |  |  |  | 28.05 | 0.895 | 0.066 |
> > > | V3 | |  ✓|  |  |  | 28.61 | 0.905 | 0.058 |
> > > | V4 | ✓|  |  | ✓  |  | 28.42 | 0.901 | 0.062 |
> > > | V5 |✓ | ✓ | ✓ |  |  | 28.92 | 0.913 | 0.052 |
> > > | V6 |✓ |✓ | ✓ | ✓ |  | 29.26 | 0.914 | 0.049 |
> > > | V7 |✓ |  ✓| ✓ | ✓ | ✓ | 29.91 | 0.919 | 0.037 |

---

> > > > ### Comment · Reviewer_cE6C · 2024-12-01
> > > >
> > > > Dear Authors,
> > > >
> > > > The feedback has successfully resolved all my concerns. As a result, I have decided to increase the score. I believe that the revisions and clarifications provided have significantly improved the quality and clarity of your work.

---

> > > > > ### Author Response · Authors · 2024-12-01
> > > > > **Response to Reviewer cE6C**
> > > > >
> > > > > Thank you for your insightful comments and appreciation of our work and rebuttal. It is good to see that our comments could address your concerns. We will do our best to improve the final version of our paper based on your valuable suggestions.

---

> ### Author Response · Authors · 2024-11-22
> **Response to Reviewer cE6C (Part 2/2)**
>
> >**Q5. Performance and Explanation Under Challenging Conditions.**
>
> Thanks for your reminder and suggestions. In Figure 2(a) of the original manuscript, we illustrate the task settings by generating low-quality images with various degradations and the corresponding initialized point clouds. These examples are solely for demonstration purposes and are not used for network training.
>
> In Figure 2(b), we visualize and render 2D images of Gaussian primitives after training on datasets constructed based on degradation settings from prior image restoration works [2, 3, 4, 5], as detailed in Section 3.1. All methods are trained under these settings for fair comparison. Our method significantly enhances rendering quality for unseen views within a certain range and maintains better performance and robustness compared to the baseline and other methods, even in challenging conditions (Figure 7, original manuscript).
>
> For extremely adverse scenarios with minimal valid information—where existing image restoration algorithms fail to recover such cases. Addressing the reconstruction of unseen views under such extreme conditions remains an open challenge for future exploration.
>
>
> ---
>
> >**Q6. Training Iterations/Time and Parameters Settings.**
>
> Thanks for your reminder. In Section 4.1 of the revised manuscript, we clarify that all 3DGS-based methods (3DGS, SRGS, and HQGS) are trained under the same settings for 50k iterations to ensure fairness. Performance comparisons based on the same number of iterations are provided in Tables 1 and 2 of the original manuscript, and comparisons based on the same training time are shown in Figure 8 of the original manuscript.
>
> Specifically, in 9 minutes, HQGS completes 35k iterations, while 3DGS completes 50k iterations, with both methods approaching near convergence. The training speeds for 3DGS, SRGS, and HQGS are 0.535 dB/kiters, 0.550 dB/kiters, and 0.824 dB/kiters, respectively. In summary, although our model has more parameters, it converges more easily and learns faster.
>
> The densification parameters are applied with a fixed number of iterations and are not learnable. For a fair comparison, we adopt the same parameters as the 3DGS pipeline.
>
>
> ---
>
> >**Q7. The Weights of MLP.**
>
> The MLP is optimized individually for each scene, as per the design established by the 3DGS pipeline [6, 7]. This means that each scene requires separate training from scratch. All comparison methods, whether they involve MLPs (such as NeRF, which uses multiple MLPs) or not, follow the same optimization approach, training each scene individually.
>
> [6] Bernhard Kerbl, Georgios Kopanas, Thomas Leimkuhler, and George Drettakis. 3D Gaussian Splatting for Real-time Radiance Field Rendering. ACM Trans.Graph, 2023, 1–14.
>
> [7] Xiang Feng, Yongbo He, Yubo Wang, Yan Yang, Wen Li, Yifei Chen, Zhenzhong Kuang, Jiajun Ding, Jianping Fan, and Jun Yu. Srgs: Super-resolution 3D Gaussian Splatting. arXiv:2404.10318, 2024

---

> ### Author Response · Authors · 2024-11-24
> **Follow-Up on Rebuttal Discussion**
>
> Dear Reviewer cE6C,
>
> We deeply appreciate your valuable feedback during the first round of review and the thoughtful discussion that has significantly helped us refine our work. Since the discussion phase ends on Nov 26, we would like to know whether we have addressed all the issues, and we would greatly welcome any additional feedback or suggestions you may have.
>
> Thank you again for your devotion to the review. If all the concerns have been successfully addressed, please consider raising the scores after this discussion phase.
>
>
> Best regards,
>
> Paper4233 Authors

---

> ### Author Response · Authors · 2024-11-26
> **Kindly Reminder for Potential Discussion**
>
> Dear Reviewer cE6C,
>
> We sincerely thank you for your valuable feedback during the first round of review and for the thoughtful discussions that have greatly contributed to improving our work. Your insights and suggestions have been instrumental in refining our submission, and we are deeply grateful for your time and effort.
>
> We kindly wish to confirm whether we have satisfactorily addressed all your concerns. Thank you again for your devotion to the review. If all the concerns have been successfully addressed, please consider raising the scores after this discussion phase.
>
> Best regards,
>
> Paper4233 Authors

---

> ### Author Response · Authors · 2024-11-30
> **Please let us know whether all issues are addressed**
>
> Dear reviewer cE6C,
>
> Thanks for the comments and review. We have provided more explanations and answers to your questions. Since the deadline for discussion is near the end, please let us know whether we have answered all the questions. Please also consider raising the score after all issues are addressed.
>
> If you have more questions, please raise them and we will reply ASAP.
>
> Thanks,
>
> Paper4233 Authors

---

### Official Review · Reviewer_rTVv · 2024-11-03

**Soundness:** 2
**Presentation:** 3
**Contribution:** 3
**Rating:** 6
**Confidence:** 4

**Summary:**

This paper presents a novel view synthesis method called HQGS, specifically optimized for low-quality images, such as those with low resolution, blur, and noise. HQGS employs an Edge-Semantic Fusion Guidance (ESFG) module to enhance the detail-capturing ability of 3D Gaussian splatting and introduces a Structural Cosine Similarity Loss (LSCS) to further improve global consistency in image rendering. Experimental results show that HQGS demonstrates stable performance across various degraded scenarios, outperforming other NeRF and 3DGS-based methods in metrics like PSNR, SSIM, and LPIPS.

**Strengths:**

1. This paper combines edge-awareness and semantic awareness through the ESFG module, providing essential high-frequency edge information to improve 3D Gaussian splatting (3DGS) reconstruction on low-quality images. The introduction of LSCS further enhances the global structural consistency of rendered images, which is an innovative design.
2. The experiments cover a wide range of common degradation conditions (e.g., low resolution, JPEG compression, blur, and noise) and compare the performance of HQGS against other state-of-the-art methods. The results demonstrate that HQGS not only outperforms these methods in image quality but also maintains efficiency in rendering time.

**Weaknesses:**

1. This approach heavily relies on high-frequency edge maps. For severely degraded images, using the Sobel operator to generate edge maps may result in significant detail loss. Given the instability of edge information in low-quality images, it is questionable whether ESFG can reliably extract edge information under various levels of degradation. There is a lack of robustness experiments on edge maps to verify the applicability of this approach.
2. The paper mentions that low-quality images produce sparse point clouds, which can negatively impact reconstruction quality. However, the paper does not clarify whether ESFG influences the density or number of Gaussian elements. If the point cloud density is insufficient, simply adjusting the distribution might not achieve optimal results.
3. Although the paper mentions that the method combines high and low-frequency information, it does not present the actual distribution of Gaussian elements in high- and low-frequency regions of the images. A lack of intuitive visualization makes it difficult to verify the practical effectiveness of ESFG and LSCS in these regions.
4. While Figure 7 demonstrates that HQGS exhibits greater robustness compared to 3DGS, it lacks a direct comparison of robustness with SRGS (e.g., in noisy or low-resolution scenarios). This omission limits the understanding of HQGS's robustness relative to other 3DGS optimization methods.
5. The paper mentions only the total training iterations but does not provide specific data on training time. Given that the addition of the ESFG module may increase training costs, the paper should ideally compare training efficiency, particularly in terms of the impact of ESFG on training duration.

**Questions:**

1. I am curious about the rationality of generating edge maps from low-quality images. Since edge maps are generated from low-quality images, can they still effectively capture key edge information in severely degraded scenes? Can the author provide edge maps with different degrees of visual degradation and the impact of failed edge map visualization on the results? Furthermore, in severely degraded scenes, is it possible to use a pre-trained image restoration model to generate high-quality images before extracting edge maps?
The paper mentions that low-quality images result in sparse point clouds but does not clarify whether the ESFG module impacts the density distribution of Gaussian elements. Can the ESFG module improve the density of the point cloud while maintaining the total number of Gaussian elements? Is there a densification strategy or explanation of how the ESFG module affects 3DGS densification to better handle the sparse point clouds generated by low-quality images?

2. The authors mention that the method combines high-frequency and low-frequency features. Could you provide a visualization of the number of the Gaussian elements across high- and low-frequency regions within an image to show how the method effectively handles these different areas?

3. Figure 7 shows only the differences between HQGS and 3DGS. Could the authors supplement this with a robustness comparison to SRGS for a more comprehensive evaluation of HQGS’s performance?

4. Could the authors provide a comparison of training times across different methods, especially discussing the impact of the ESFG module on training time?

---

> ### Author Response · Authors · 2024-11-22
> **Response to Reviewer rTVv (Part 2/2)**
>
> >**Q4. Robustness Validation on all methods.**
>
> As suggested, we provide the results of other methods in the following Table and Table 5 of the revised manuscript. Our method performs favorably against others and exhibits stronger generalization ability under challenging conditions.
>
> |        | **Gaussian noise** 0      | **Gaussian noise** 10     | **Gaussian noise** 25    | **Gaussian noise** 50    | **Low resolution** 1×    | **Low resolution** 2×    | **Low resolution** 4×    | **Low resolution** 8×    |
> |-|-|-|-|-|-|-|-|-|
> |   **Methods** | **PSNR↑** | **LPIPS↓**     | **PSNR↑** | **LPIPS↓**     | **PSNR↑** | **LPIPS↓**     | **PSNR↑** | **LPIPS↓**     | **PSNR↑** | **LPIPS↓**     | **PSNR↑** | **LPIPS↓**     | **PSNR↑** | **LPIPS↓**     |
> | **NeRF**            | 30.42     | 0.072         | 29.11     | 0.079         | 27.78     | 0.091         | 23.04     | 0.143         | 29.83     | 0.113         | 29.71     | 0.128         | 28.83     | 0.138         | 28.16     | 0.148         |
> | **3DGS**      | 30.21     | 0.043         | 29.49     | 0.052         | 27.46     | 0.077         | 23.05     | 0.109         | 31.26     | 0.091         | 30.18     | 0.094         | 29.25     | 0.092         | 28.63     | 0.111         |
> | **NeRFLix**         | 30.86     | 0.054         | 29.47     | 0.063         | 27.04     | 0.071         | 23.12     | 0.112         | 31.42     | 0.069         | 30.87     | 0.069         | 30.12     | 0.076         | 29.66     | 0.096         |
> | **SRGS**            | 30.71     | 0.036         | 29.17     | 0.045         | 27.63     | 0.061         | 23.03     | 0.126         | 31.36     | 0.078         | 30.93     | 0.065         | 30.54     | 0.061         | 30.11     | 0.087         |
> | **HQGS (Ours)**     | **31.32** | **0.018**     | **30.41** | **0.029**     | **28.63** | **0.043**     | **26.31** | **0.067**     | **32.08** | **0.031**     | **31.85** | **0.033**     | **31.61** | **0.038**     | **31.37** | **0.051**     |
>
> ---
>
> >**Q5. Training Time vs Performance.**
>
> Figure 8 in the original manuscript compares existing methods under the same training time for fairness. HQGS demonstrates a faster convergence rate and consistently performs well against other methods, highlighting its training efficiency. Whether compared under the same training time or at convergence with the same number of iterations, our method shows better performance.
>
> To improve clarity, we have updated the subsection title from "ANALYSIS ON RECONSTRUCTION TIME OF SOME 3DGS-BASED METHODS." to "TRAINING TIME VS QUALITY." We have also improved the readability of Figure 8 in the revised manuscript to eliminate overlapping issues.
>
> ---
>
> >**Q6. The Decline in Edge Detection Performance and Its Impact After Applying Image Restoration Processing.**
>
> Figure 7 of the original manuscript validates this claim: under challenging conditions (e.g., noise level 50 and 8 $\times$ downsampling), edge detection performance declines compared to the clean case, leading to a drop in the quality of rendered unseen views. Despite this, our method still performs better, achieving a 3.27 dB improvement over 3DGS at a noise level 50.
>
> Additionally, as shown in the Table in Q4, our HQGS remains superior even in clean scenes (noise = 0, 1 $\times$ resolution), which can be considered the upper limit after image restoration. These results demonstrate that our method is effective not only under challenging conditions but also in clean settings, excelling in capturing small objects and enhancing global learning in low-frequency regions.
>
> ---
>
> >**Q7. Can the ESFG module improve the density of the point cloud while maintaining the total number of Gaussian elements? Is there a densification strategy or explanation of how the ESFG module affects 3DGS densification to better handle the sparse point clouds generated by low-quality images?**
>
> In 3DGS, once training concludes, the points in the point cloud correspond directly to the centers of Gaussian primitives [6, 7], meaning their quantities are fixed and equal, cannot be independently adjusted.
>
> Regarding density, the original densification strategy in 3DGS duplicates points based on factors like positional importance or the presence of details. However, 3DGS exhibits limited sensitivity to fine details, leading to missing object features (e.g., the "wires" in Figure 2(b) of the original manuscript). In contrast, our ESFG emphasizes finer details and provides this information to the model, resulting in a higher density of Gaussian primitives, specifically in detailed regions, and effectively capturing these intricate features.
>
> [6] Bernhard Kerbl, Georgios Kopanas, Thomas Leimkuhler, and George Drettakis. 3D Gaussian Splatting for Real-time Radiance Field Rendering. ACM Trans.Graph, 2023: 1–14.
>
> [7] Xiang Feng, Yongbo He, Yubo Wang, Yan Yang, Wen Li, Yifei Chen, Zhenzhong Kuang, Jiajun Ding, Jianping Fan, and Jun Yu. SRGS: Super-resolution 3D Gaussian Splatting. arXiv:2404.10318, 2024

---

> ### Author Response · Authors · 2024-11-22
> **Response to Reviewer rTVv (Part 1/2)**
>
> We thank reviewer rTVv for acknowledging the contribution of our paper and providing thoughtful comments.
>
>
> >**Q1. Lack of Robustness Experiments on Edge Maps**
>
>
> We leverage the well-known fact that edge maps have sufficient frequency information [1] and can be obtained by an edge detection operator even from degraded images. As suggested, we visualize edge detection results obtained from the Sobel operator under progressively challenging conditions in Figure 1 of the submitted Supplementary Material. Additionally, we compute the PSNR values based on images and edge maps under various conditions with respect to clean ones, then present the PSNR variance of the images and edge maps: 10.18/0.285. These results further demonstrate that the edge maps are more robust under progressively degraded conditions. Thus, edge detection performs well in extracting edge information to a considerable extent, even under relatively challenging conditions.
>
> In our experimental setups, we follow the degradation settings commonly used in image restoration tasks [2, 3, 4, 5], ensuring that the chosen degradation ranges align with existing works (4$\times$ downsampling, JPEG with quality level 10, and so on). Under these settings, our method consistently performs favorably against existing algorithms. Furthermore, we discuss the progressively challenging scenarios in Figure 7 of the original manuscript, showing that our method performs better than other approaches under severe conditions (e.g., noise level of 50 or 8 $\times$ downsampling). We include comparisons with more methods in Table 6 of the revised manuscript.
>
> [1] Lindeberg T. Scale space. Encyclopedia of Computer Science and Engineering. 2009: 2495–2504.
>
> [2] Li Y, Fan Y, Xiang X, et al. Efficient and explicit modelling of image hierarchies for image restoration. CVPR. 2023: 18278-18289.
>
> [3] Ren B, Li Y, Mehta N, et al. The ninth NTIRE 2024 efficient super-resolution challenge report. CVPR. 2024: 6595-6631.
>
> [4] Lu Z, Li J, Liu H, et al. Transformer for single image super-resolution. CVPR. 2022: 457-466.
>
> [5] El Helou M, Süsstrunk S. Blind universal Bayesian image denoising with Gaussian noise level learning. TIP. 2020, 29: 4885-4897.
>
> ---
>
> >**Q2. The Impact of ESFG on Gaussian Primitives Density.**
>
> Figure 2(b) in the original manuscript demonstrates the centers of Gaussian primitives from the trained model, which are also the points in the point cloud. Compared to 3DGS, our method produces a denser distribution of Gaussian primitives, enabling enhanced coverage of finer details and textures rather than only redistributing the primitives.
>
> ---
>
> >**Q3. Visualizations of Gaussian Primitives in high- and low-frequency regions.**
>
>
> The proposed ESFG is designed to guide the model in focusing on detailed regions and small objects, as validated by Figure 2(b) in the original manuscript. Following your suggestion, we provide additional evidence in Figure 2 of the submitted Supplementary Material.
>
> Specifically, we illustrate the Gaussian primitives in both low- and high-frequency regions, alongside their corresponding rendering results. To highlight these regions, we adjust the point cloud's angle, select optimal viewpoints, and include enlarged screenshots. Our method demonstrates better visual quality, particularly in areas like the floor and windows.
>
>
> Additionally, we show the difference map between rendered and clean images in Figure 3 of the Supplementary Material (Figure 6 in the revised manuscript). These results show that 3DGS and other methods produce significant inaccuracies in low- and high-frequency regions (highlighted as bright areas in the difference maps). In contrast, our method achieves smaller differences, demonstrating well performance in these challenging regions.

---

> ### Author Response · Authors · 2024-11-24
> **Follow-Up on Rebuttal Discussion**
>
> Dear Reviewer rTVv,
>
> We deeply appreciate your valuable feedback during the first round of review and the thoughtful discussion that has significantly helped us refine our work. Since the discussion phase ends on Nov 26, we would like to know whether we have addressed all the issues, and we would greatly welcome any additional feedback or suggestions you may have.
>
> Thank you again for your devotion to the review. If all the concerns have been successfully addressed, please consider raising the scores after this discussion phase.
>
>
> Best regards,
>
> Paper4233 Authors

---

> ### Author Response · Authors · 2024-11-26
> **Kindly Reminder for Potential Discussion**
>
> Dear Reviewer rTVv,
>
> We sincerely thank you for your valuable feedback during the first round of review and for the thoughtful discussions that have greatly contributed to improving our work. Your insights and suggestions have been instrumental in refining our submission, and we are deeply grateful for your time and effort.
>
> We kindly wish to confirm whether we have satisfactorily addressed all your concerns. Thank you again for your devotion to the review. If all the concerns have been successfully addressed, please consider raising the scores after this discussion phase.
>
> Best regards,
>
> Paper4233 Authors

---

> ### Comment · Reviewer_rTVv · 2024-11-26
> **Official Comment by Reviewer rTVv**
>
> Dear Authors,
>
> Thank you for your detailed and thorough response. I appreciate the effort you put into addressing the concerns raised during the review process, as well as for providing additional data and visualizations.
>
> However, I still have some unresolved questions regarding the ESFG module and its impact on the densification of Gaussian primitives in HQGS. Specifically, I have observed that HQGS consistently produces more Gaussian primitives compared to 3DGS, both in high-frequency regions and in low-frequency areas. This observation raises the following points:
>
> 1. Are you using the same densification strategy as the original 3DGS method? If so, what mechanism within the ESFG module leads to the observed increase in Gaussian primitives across both high- and low-frequency regions?
> 2. If the densification frequency or strategy has been intentionally adjusted in HQGS, could you provide more details on how it differs from the original 3DGS approach? Specifically, has the frequency of densification been increased, or has a new mechanism been introduced to achieve this result?
>
> Clarifying these points would provide valuable insights into how HQGS operates and the role of ESFG in improving reconstruction quality. I look forward to your response.
>
> Reviewer rTVv

---

> ### Author Response · Authors · 2024-11-27
> **Response to Reviewer rTVv**
>
> Thank you for your thoughtful feedback and for taking the time to review our rebuttal. We will address your points regarding the proposed modules below and include these in the revised version to improve its readability.
>
>
> >**1. Are you using the same densification strategy as the original 3DGS method? If so, what mechanism within the ESFG module leads to the observed increase in Gaussian primitives across both high- and low-frequency regions?**
>
> We adopt the same densification strategy as the other 3DGS-based methods [1,2], performing a densification operation every 100 iterations to expand the point cloud density in important regions. The observed increase in Gaussian primitives is due to the introduction of additional information that guides the model to areas where point cloud expansion is necessary.
>
> For the ESFG module, we aim to improve attention to these areas by incorporating a high-frequency map. The high-frequency branch contains only high-frequency information, which has superior perceptual ability for small target regions. Additionally, we introduce the original low-quality images with strong semantic information, which include both high- and low-frequency area information and serve as supplementary input. Furthermore, the proposed SCS loss function can also be used to supplement low-frequency information.
>
> Our ablation studies in Tables 3 and 4 of the revised paper validate these previous points. Moreover, as seen in Figure 2 of the supplementary materials, the distribution of Gaussian primitives indicates that our method generates more Gaussian primitives in both high-frequency regions (e.g., the dinosaur) and low-frequency regions (e.g., the floor), compared to 3DGS. Due to the ESFG module, the number of Gaussian primitives in high-frequency regions increases significantly.
>
>
> ---
>
> >**2. If the densification frequency or strategy has been intentionally adjusted in HQGS, could you provide more details on how it differs from the original 3DGS approach? Specifically, has the frequency of densification been increased, or has a new mechanism been introduced to achieve this result?**
>
> We do not increase the densification frequency; all of our parameter settings are consistent with other 3DGS-based methods [1,2] to ensure fairness. Densification occurs once every 100 iterations. We believe the increase in the number of Gaussian primitives is primarily due to the enhanced attention information. Since the densification operation is performed based on the importance of the information, it only occurs in regions where the model determines that higher distinction is needed, thereby improving the accuracy of the model's rendering results.
>
> The ESFG module provides sufficient guidance (as mentioned in Question 1), directing the model to focus on specific regions. This helps ensure that more Gaussian primitives are allocated to fit the ground truth, thereby reducing the loss.
>
> [1] Bernhard Kerbl, Georgios Kopanas, Thomas Leimkuhler, and George Drettakis. 3D Gaussian Splatting for Real-time Radiance Field Rendering. ACM Trans.Graph, 2023: 1–14.
>
> [2] Xiang Feng, Yongbo He, Yubo Wang, Yan Yang, Wen Li, Yifei Chen, Zhenzhong Kuang, Jiajun Ding, Jianping Fan, and Jun Yu. SRGS: Super-resolution 3D Gaussian Splatting. arXiv:2404.10318, 2024

---

> ### Author Response · Authors · 2024-11-30
> **Please let us know whether all issues are addressed**
>
> Dear reviewer rTVv,
>
> Thanks for the comments and review. We have provided more explanations and answers to your questions. Since the deadline for discussion is near the end, please let us know whether we have answered all the questions. Please also consider raising the score after all issues are addressed.
>
> If you have more questions, please raise them and we will reply ASAP.
>
> Thanks,
>
> Paper4233 Authors

---

> ### Author Response · Authors · 2024-12-02
> **Follow-Up on Rebuttal Discussion**
>
> Dear Reviewer rTVv,
>
> Thank you once again for your insightful feedback. With the deadline approaching on December 2, we would greatly appreciate the opportunity to clarify any remaining concerns or answer any questions you may have.
>
> If all issues have been addressed to your satisfaction, we kindly ask you to consider revising the scores accordingly after this discussion phase. We look forward to your continued feedback and hope to resolve any lingering doubts as efficiently as possible.
>
> Thank you again for your time and dedication to this review!
>
> Best,
>
> Paper4233 Authors

---

> ### Comment · Reviewer_rTVv · 2024-12-02
> **Official Comment by Reviewer rTVv**
>
> Thank you for your clear and detailed responses. Your explanations have addressed my concerns thoroughly, and I will increase the scores.

---

> > ### Author Response · Authors · 2024-12-02
> > **Response to Reviewer rTVv**
> >
> > Thank you for your insightful comments and appreciation of our work and rebuttal. It is good to see that our comments could address your concerns. We will do our best to improve the final version of our paper based on your valuable suggestions.

---

### Author Response · Authors · 2024-11-22
**Summary**

We thank all reviewers for their positive feedback:
* The proposed method and designed modules are innovative and good (rTVv, cE6C, s4XL, and enTq).
* SOTA performance with superior robustness and maintains efficiency in rendering time (rTVv, cE6C, s4XL, and enTq).
* Sufficient experiments (rTVv, s4XL). Well writen (s4XL).
* The ablation studies are quite thorough (s4XL).

We address the raised concerns as follows.

---

### Comment · Area_Chair_WWtM · 2024-11-24

Dear Reviewers,

The discussion with the authors will conclude soon. The authors have provided detailed rebuttals. If there are any points that you feel have not been adequately clarified or if there are misunderstandings in their responses, please take this opportunity to raise them now. Thank you for your contributions to this review process.

---

### Meta-Review · Area_Chair_WWtM · 2024-12-21

**Metareview:**

The paper introduces HQGS, a view synthesis method optimised for low-quality images (e.g., low resolution, blur, noise). It combines an Edge-Semantic Fusion Guidance (ESFG) module to improve detail capture in 3D Gaussian splatting with a Structural Cosine Similarity Loss (LSCS) for enhanced global structural consistency. Experiments demonstrate HQGS's superior performance across various degraded scenarios, outperforming state-of-the-art methods in PSNR, SSIM, and LPIPS metrics.

Reviewers praised the novel ESFG module and LSCS for effectively addressing challenges in degraded images.  ESFG module effectively integrates high-frequency edge information into 3D Gaussian splatting while LSCS enhances global structural consistency in rendered images.
Comprehensive experiments covering diverse degradation conditions.

HQGS introduces innovations, with strong experimental validation and unanimous reviewer support. It is a significant contribution to view synthesis for degraded images. Therefore, I recommend an acceptance.

**Additional Comments On Reviewer Discussion:**

Concerns about clarifying ESFG integration and providing more comparisons were addressed in the rebuttal, leading to unanimous acceptance.

---

### Decision · Program_Chairs · 2025-01-22

Accept (Poster)